# Loss of $N^1$-methylation of G37 in tRNA induces ribosome stalling and reprograms gene expression

Isao Masuda[1†], Jae-Yeon Hwang[2†], Thomas Christian[1†], Sunita Maharjan[1], Fuad Mohammad[2], Howard Gamper[1], Allen R Buskirk[2]*, Ya-Ming Hou[1]*

[1]Department of Biochemistry and Molecular Biology, Thomas Jefferson University, Philadelphia, United States; [2]Department of Molecular Biology and Genetics, Johns Hopkins University School of Medicine, Baltimore, United States

**Abstract** $N^1$-methylation of G37 is required for a subset of tRNAs to maintain the translational reading-frame. While loss of m¹G37 increases ribosomal +1 frameshifting, whether it incurs additional translational defects is unknown. Here, we address this question by applying ribosome profiling to gain a genome-wide view of the effects of m¹G37 deficiency on protein synthesis. Using *E coli* as a model, we show that m¹G37 deficiency induces ribosome stalling at codons that are normally translated by m¹G37-containing tRNAs. Stalling occurs during decoding of affected codons at the ribosomal A site, indicating a distinct mechanism than that of +1 frameshifting, which occurs after the affected codons leave the A site. Enzyme- and cell-based assays show that m¹G37 deficiency reduces tRNA aminoacylation and in some cases peptide-bond formation. We observe changes of gene expression in m¹G37 deficiency similar to those in the stringent response that is typically induced by deficiency of amino acids. This work demonstrates a previously unrecognized function of m¹G37 that emphasizes its role throughout the entire elongation cycle of protein synthesis, providing new insight into its essentiality for bacterial growth and survival.

*For correspondence:
buskirk@jhmi.edu (ARB);
ya-ming.hou@jefferson.edu (Y-MH)

[†]These authors contributed equally to this work

**Competing interests:** The authors declare that no competing interests exist.

## Introduction

$N^1$-methylation of G37 in tRNA, generating m¹G37 on the 3'-side of the anticodon, is a post-transcriptional modification that is essential for life (*Björk et al., 2001*). It has been specifically associated across evolution with all isoacceptors of tRNA^Pro, which are species that share the same prolyl specificity of aminoacylation but differ in the primary sequence and in the anticodon triplet and yet collectively decode the Pro CCN codons (N = A, C, G, and U). Similarly, m¹G37 is conserved in the CCG isoacceptor of tRNA^Arg(tRNA^Arg(CCG)), with the anticodon CCG for pairing with the CGG codon, and it is conserved in isoacceptors tRNA^Leu(GAG), tRNA^Leu(UAG), and tRNA^Leu(CAG) for pairing with CUU and CUC codons (CU[C/U]) and CU[A/G] codons, respectively (*Björk et al., 2001*; *Li et al., 1997*). Additionally, m¹G37 may also be present in other tRNAs in higher eukaryotes. The m¹G37 methylation of tRNA is required for cell growth and viability; elimination of the enzyme responsible for the methylation causes cell death in yeast and in bacteria (*Björk et al., 2007*; *Björk et al., 2001*; *Masuda et al., 2019*). The established function of m¹G37 is to maintain the translational reading-frame during protein synthesis (*Björk et al., 1989*; *Hagervall et al., 1993*; *Qian et al., 1998*). Loss of m¹G37 elevates frequencies of ribosomal +1 frameshifting in kinetic assays with reconstituted *E. coli* ribosomes (*Gamper et al., 2015a*), and in cell-based assays in *E. coli* and *Salmonella* (*Gamper et al., 2021*; *Gamper et al., 2015a*). Unlike ribosomal miscoding, +1 frameshifting is almost always deleterious, altering the translational reading-frame, inducing premature termination of protein synthesis, and ultimately cell death.

Recent work has shed light on how loss of m$^1$G37 induces ribosomal +1 frameshifting. While non-methylated tRNAs were thought to pair incorrectly in the ribosomal A site (the aminoacyl-tRNA [aa-tRNA] binding site) during decoding (*Roth, 1981*), they were found to occupy the correct reading-frame in X-ray crystal structures and in kinetic assays (*Gamper et al., 2021*; *Maehigashi et al., 2014*). Maintenance of the correct reading-frame in the A site is consistent with the strict ribosomal A site structure that selects for accurate anticodon-codon pairing during decoding (*Ogle et al., 2001*; *Ogle and Ramakrishnan, 2005*). Even genetically isolated high-efficiency +1–frameshifting tRNAs, which usually contain an extra nucleotide inserted to the anticodon loop (*Atkins and Björk, 2009*), were found to occupy the correct reading-frame in the A site (*Dunham et al., 2007*; *Fagan et al., 2014*; *Gamper et al., 2021*; *Maehigashi et al., 2014*). These genetically isolated +1-frameshifting tRNAs, however, were found to occupy the triplet +1-frame in the ribosomal P site (the peptidyl-tRNA binding site) (*Hong et al., 2018*).

All existing evidence points to tRNA +1 frameshifting occurring after decoding at the A site. A frameshift-prone tRNA, containing the natural anticodon loop but lacking m$^1$G37, is shown in kinetic assays to undergo +1 frameshifting during translocation from the A site to the P site, or during occupancy in the P site next to an empty A site (*Gamper et al., 2015a*). A genetically isolated high-efficiency +1-frameshifting tRNA, containing an expanded anticodon loop, is also shown in kinetic assays to undergo +1 frameshifting during translocation (*Gamper et al., 2021*). The notion of +1 frameshifting during translocation is supported by a recent cryo-EM structural analysis of a canonical tRNA translating a frameshift-prone mRNA sequence (*Demo et al., 2021*). Similarly, the notion of +1 frameshifting within the P site is supported by X-ray crystal structures of a P site-bound frameshift-prone tRNA with a natural anticodon loop (*Hoffer et al., 2020*). In these latter structures, while m$^1$G37 stabilizes the tRNA on an mRNA codon that is prone to frameshifting, loss of m$^1$G37 destabilizes the tRNA-ribosome interaction and induces +1 frameshifting within large conformational changes of the ribosome.

Here, we seek to determine whether m$^1$G37 plays additional roles beyond maintaining the ribosome translational reading-frame. An earlier study suggested that loss of m$^1$G37 delayed the tRNA anticodon-codon pairing interaction during decoding (*Li et al., 1997*), raising the possibility of a role at the ribosomal A site. To test this possibility more broadly, we employ the approach of ribosome profiling to determine ribosome positions during translation of the entire transcriptome of a cell (*Ingolia et al., 2009*) and to monitor how ribosome density changes upon loss of m$^1$G37. Using *E. coli* as a model, we show that deficiency of m$^1$G37 induces global ribosome stalling, most notably at Pro codons CCN, the Arg codon CGG, and the Leu codon CUA, all of which are translated by tRNAs that are normally methylated with m$^1$G37. Stalling is most prominent when the affected codons are being decoded at the ribosomal A site, indicating a distinct mechanism than that of +1 frameshifting. Enzyme- and cell-based assays show that m$^1$G37 deficiency reduces aminoacylation of the affected tRNAs, including all isoacceptors of tRNA$^{Pro}$ and the isoacceptor tRNA$^{Arg}$(CCG), and that additionally it reduces rates of peptide-bond formation for some of these tRNAs. Most significantly, stalling induces programmatic changes in gene expression that are consistent with changes occurring during the bacterial stringent response under environmental stress of nutrient starvation. These findings support a model in which m$^1$G37 deficiency reduces levels of aa-tRNAs at the ribosomal A site and prevents peptide-bond formation in some cases, leading to ribosome stalling. Binding of uncharged tRNAs to the A site would then induce programmatic changes of gene expression similar to those occurring during the bacterial stringent response through activation of the ppGpp synthase RelA (*Gourse et al., 2018*). The importance of m$^1$G37 for the ribosomal activity at the A site, together with its already demonstrated importance in maintaining the translational reading-frame from the A site to the P site and within the P site (*Gamper et al., 2021*; *Gamper et al., 2015a*), establishes the involvement of the methylation throughout the entire elongation cycle of protein synthesis. This sustained involvement of m$^1$G37 during protein synthesis underscores its indispensable role in bacterial viability and survival.

## Results

### *E. coli* strains with conditional m$^1$G37 deficiency

Because m$^1$G37 is essential for cell viability, a simple knock-out of the gene responsible for its biosynthesis cannot be made. Previous studies of cellular functions of m$^1$G37 relied on temperature-sensitive variants of the gene responsible for m$^1$G37 biosynthesis whose protein product became inactivated at elevated temperatures (*Björk and Nilsson, 2003*; *Masuda et al., 2013*). Because elevated temperatures induce changes in gene expression, we took a different approach to conditionally deplete m$^1$G37 to study its role in protein synthesis. Interestingly, while m$^1$G37 is conserved in evolution, the genes responsible for its biosynthesis are distinct – being *trmD* in bacteria and *trm5* in archaea and eukaryotes (*Christian et al., 2004*). The protein products of *trmD* and *trm5* are fundamentally different from each other in structure and mechanism (*Christian and Hou, 2007*; *Christian et al., 2010a*; *Christian et al., 2010b*; *Christian et al., 2016*; *Lahoud et al., 2011*; *Sakaguchi et al., 2014*). We recently constructed conditional m$^1$G37-deficient strains of *E. coli* and *Salmonella*, in which the *trmD* locus is deleted from the chromosome and cell viability of the *trmD-knock-out* (*trmD-KO*) strain is maintained by a plasmid-borne human *trm5* that is under arabinose (Ara)-controlled expression (*Gamper et al., 2015a*; *Masuda et al., 2019*). Upon induction with Ara, expression of human *trm5* is sufficient to supply m$^1$G37-tRNAs to support bacterial viability (*trmD-KO* (*trm5+*)) (*Christian et al., 2004*), whereas upon replacement of Ara with glucose (Glc), expression of human *trm5* is arrested and the human enzyme is degraded inside bacterial cells (*trmD-KO* (*trm5–*)) (*Christian et al., 2013*). As a control, a *trmD-wild-type* (*trmD-WT*) strain was created, where *trmD* remains on the chromosome and expression of the plasmid-borne *trm5* in the presence of Ara (*trmD-WT* (*trm5+*)), or its repression in the presence of Glc (*trmD-WT* (*trm5–*)), did not affect cell viability.

To avoid the possibility of artifacts by studying only one conditional m$^1$G37-deficient strain, we developed a second conditional m$^1$G37-deficient strain to compare data and to strengthen conclusions. In this second conditional strain, we chose *E. coli* as a model and extended *trmD* at the chromosomal locus with a degron sequence, adding YALAA to the C-terminus of the TrmD protein (the *trmD-deg* strain) to allow rapid degradation by the protease ClpXP. We controlled TrmD degradation by inducing the expression of *clpXP* from a plasmid using Ara, or by repressing the expression using Glc (*Carr et al., 2012*). Because over-expression of the plasmid-borne *clpXP* could also target proteins without the degron tag, we generated a control strain (*trmD-cont*) for comparison. In this *trmD-cont* strain, we added the coding sequence for the degron tag after the stop codon of *trmD* to maintain the same gene length as in *trmD-deg*, but without expression of the tag (*Figure 1—figure supplement 1A and B*).

Overnight cultures of *trmD-deg* and *trmD-cont* strains were grown in LB + Glc and were spotted onto an LB + Ara plate to turn on expression of *clpXP*. Analysis of a serial dilution of each strain confirmed that *trmD-deg* cells rapidly lost viability, whereas *trmD-cont* cells retained viability (*Figure 1A*). In liquid culture, in which each strain was freshly diluted into LB + Ara at OD$_{600}$ 0.1 and grown to 0.3–0.4, followed by a second round of dilution and re-growth, we observed that the re-growth of the *trmD-deg* strain was retarded in the third round, whereas that of the *trmD-cont* strain was robust (*Figure 1B*). Although the growth defect of the *trmD-deg* strain only manifested in the third cycle of dilution, the level of TrmD protein drastically decreased in 30 min after the first dilution into fresh LB + Ara (as in *Figure 1B*), whereas that in the *trmD-cont* strain remained stable up to 90 min (*Figure 1C*) and longer (*Figure 1—figure supplement 1C*). We speculate that it takes time for cultures to express *clpXP* to a threshold level, after which TrmD is rapidly degraded. A similar pattern of drastic reduction of the target protein following *clpXP* induction was reported previously (*Carr et al., 2012*). We also speculate that the *clpXP* expression plasmid, which has the strongest expression strength among a library of plasmids varying in Shine-Dalgarno sequences (Materials and methods), could also contribute to the drastic degradation of TrmD.

To measure cellular levels of m$^1$G37 during this series of dilution and re-growth, we designed a 5'-[$^{32}$P]-labeled primer of 14 nucleotides (14 nt) complementary to *E. coli* tRNA$^{Leu}$(CAG). The presence of m$^1$G37 would inhibit primer extension, producing a 15 nt product, whereas the absence of m$^1$G37 would permit primer extension to the 5'-end, producing a 53 nt product. The level of m$^1$G37 was calculated as the fraction of the 15 nt product in the sum of all of the primer-extension products of each reaction. The primer was not included in the calculation due to its molar excess in these

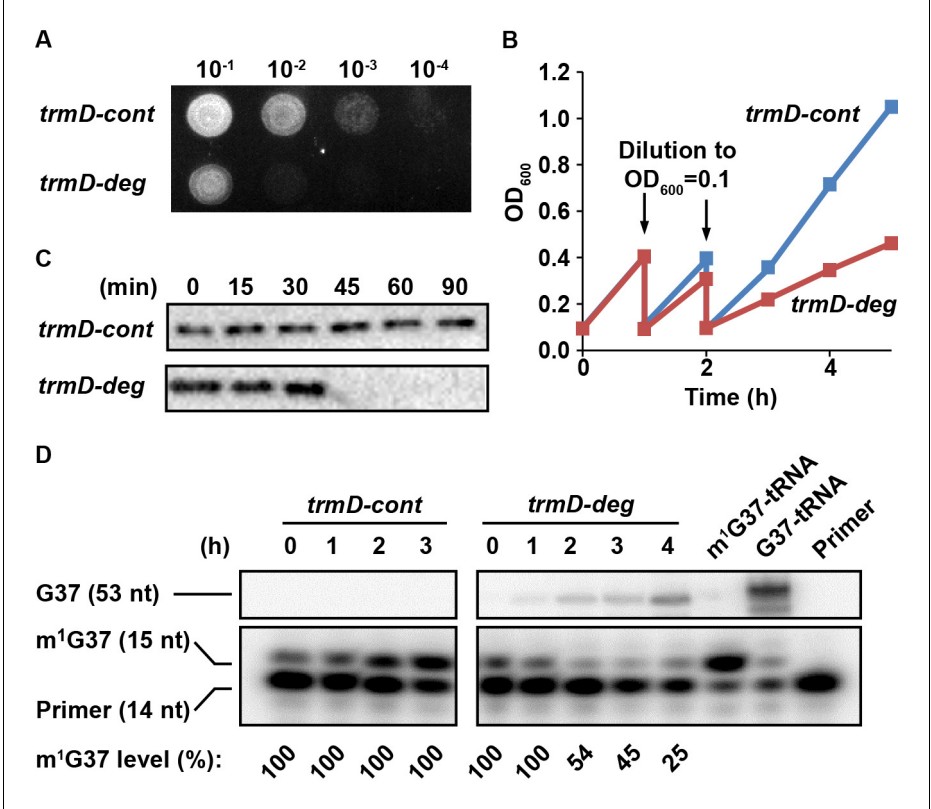

**Figure 1.** Conditional m$^1$G37 deficiency of the *E. coli trmD-deg* strain. (**A**) Growth of *trmD-deg* and *trmD-cont* G78 cells on an LB plate. An overnight culture of *trmD-deg* and *trmD-cont* cells in LB + Glc was serially diluted, spotted on an LB + Ara plate to turn on *clpXP* expression, and incubated at 37°C overnight. (**B**) Representative growth of *trmD-deg* and *trmD-cont* G78 cells in a liquid LB culture. An overnight culture of *trmD-deg* and *trmD-cont* G78 cells in LB + Glc was grown in LB for 1–2 hr, then diluted into LB + Ara at OD$_{600}$ of 0.1 and grown to OD$_{600}$ of 0.3 at 37°C. The cycle of dilution and re-growth was repeated three times to observe a growth defect of the *trmD-deg* strain. (**C**) Western blot analysis of TrmD protein. An overnight culture of *trmD-deg* and *trmD-cont* cells in LB + Glc was diluted to LB + Ara at T = 0, and was sampled at the indicated time points. Cell lysates were separated on a 12% SDS-PAGE and TrmD protein was detected by primary antibody against *E. coli* TrmD and a secondary antibody against rabbit IgG. (**D**) Primer-extension inhibition analysis. An overnight culture of *trmD-deg* and *trmD-cont* cells in LB + Glc was diluted to LB + Ara at T = 0 and the fresh culture was taken through three cycles of dilution and re-growth. Total RNA was extracted over the time course, probed with a 5′-[$^{32}$P]-labeled DNA primer targeting *E. coli* tRNA$^{Leu/CAG}$, and analyzed by a 12% PAGE/7 M urea gel and phosphorimaging. Primer extension would terminate in the control at 1 nt downstream of m$^1$G37, generating a 15 nt fragment, whereas primer extension would continue to the 5′-end in TrmD deficiency, generating a 53 nt fragment. The fraction of m$^1$G37 in each sample is calculated based on all primer-extension products as shown in the source file, not including the primer. Because the same amount of cell culture was used for extraction of RNA, an increased primer-extension stop at the 15 nt band relative to the primer position was observed in *trmD-cont* cells, reflecting an increased cell density 0–3 hr and increased synthesis of m$^1$G37-tRNA. In all samples collected for *trmD-cont* cells, no synthesis of the read-through 53 nt band was observed, indicating 100% methylation. At the time of cell harvest, the m$^1$G37 level was 100% for *trmD-cont* cells at T = 3 hr, but was below 25% for *trmD-deg* cells at T = 4 hr.

The online version of this article includes the following figure supplement(s) for figure 1:

**Figure supplement 1.** Genetic construction and validation of *E. coli trmD-deg* and *trmD-cont* strains.

reactions. Analysis of total RNA samples, collected after the first dilution into fresh LB + Ara in the time course (as in *Figure 1B*), showed that the 15 nt product progressively decreased in *trmD-deg* cells, indicating gradual loss of m$^1$G37, but that it remained stable in *trmD-cont* cells (*Figure 1D*). At the time of cell harvesting, m$^1$G37 in *trmD-deg* cells was at or below 25% (T = 4.2 hr), whereas that in *trmD-cont* cells was 100% (T = 3 hr). Collectively, these results demonstrate the ability of the

degron approach to control the stability of TrmD and to produce conditional m$^1$G37 deficiency in *E. coli* cells.

## Codon-specific ribosome stalling in m$^1$G37 deficiency

We performed ribosome profiling and RNA-seq analyses on two biological replicates of the *trmD-deg* and *trmD-cont* strains, which were cultured in the presence of *clpXP* expression with three cycles of dilution and re-growth and were harvested at OD$_{600}$ of 0.3. We also obtained a third set of samples using *trmD-KO* (*trm5*–) and *trmD-WT* (*trm5*–) strains, which were cultured in the absence of *trm5* expression in three cycles of dilution and re-growth and were harvested at OD$_{600}$ of 0.3. Given that m$^1$G37 is associated with a specific set of *E. coli* tRNAs, we looked in the ribosome profiling data for local differences in the A site occupancy for each codon in m$^1$G37 deficiency. If loss of m$^1$G37 impaired tRNA decoding, we expected that the ribosome would linger on affected codons to accumulate higher levels of density. To quantify these codon-specific effects, we defined a pause score for each codon as the ribosome density at the first nt of the codon normalized by the average ribosome density on the gene where that codon occurs. We computed the pause score for all of the 61 sense codons by averaging the pause score at thousands of instances of each codon across the entire transcriptome. Strikingly, we observed significant increases in the pause score for a set of codons when each was positioned at the ribosomal A site during decoding (*Figure 2*). Most notably, ribosome density at Pro codons was dramatically increased in m$^1$G37 deficiency, showing an increase in the pause score from 1.4 in *trmD-cont* cells to 3.5 in *trmD-deg* cells (*Figure 2A*, left). Similarly, the pause score at Pro codons increased from 1.2 in *trmD-WT* (*trm5*–) cells to 2.5 in *trmD-KO* (*trm5*–) cells (*Figure 2A*, right). The increase in the pause score for all Pro codons indicates that m$^1$G37 deficiency affected decoding by some or all of the isoacceptors of tRNA$^{Pro}$, leading to strong ribosome pausing during elongation.

We also observed differences in ribosome pausing at other codons (*Figure 2A*), which likely resulted from artifacts of the ribosome profiling method and not from translation differences due to m$^1$G37 deficiency. There were stronger pauses at Ser and Gly codons in the control samples of *trmD-cont* than in the *trmD-deg* samples, and stronger pauses at Ser codons in the control samples of *trmD-WT* (*trm5*–) than in the *trmD-KO* (*trm5*–) samples. These pauses were likely due to artifacts of harvesting bacterial cultures. We previously observed pauses at Ser and Gly codons in cells harvested by filtration and showed that filtration reduced aminoacylation of the tRNA cognate to these codons (*Mohammad et al., 2019*). These Ser and Gly pauses, however, are less obvious in strains where protein synthesis is defective (e.g., due to loss of the elongation factor EFP) (*Woolstenhulme et al., 2015*), which may explain their lower pause scores in the *trmD-deg* and *trmD-KO* (*trm5*–) samples than in the control samples. We have also observed variable pausing at Thr codons under different harvesting conditions in unpublished work, although those pauses have not been as well characterized. Given that all of the samples described here were harvested by filtration, we attributed the pauses at Ser, Gly, and Thr codons (labeled in blue in *Figure 2A*) to artifacts of cell growth and harvesting and did not consider them further.

At the codon level, we observed higher pause scores for all four Pro codons (CCN, in red, *Figure 2B*) in m$^1$G37-deficient cells relative to control cells, as shown in *trmD-deg* vs. *trmD-cont* strains (left) and in *trmD-KO* (*trm5*–) vs. *trmD-WT* (*trm5*–) strains (right). In both replicates of the *trmD-deg* strain, Pro codons CCG and CCA had significantly higher pause scores than CCC and CCU. In contrast, all four CCN codons showed similar pause scores in the *trmD-KO* (*trm5*–) strain. While the reason for these differences between the two strains is not clear, the implication is that ribosomes paused at all four Pro codons in both strains. Also, we observed elevated pause scores on the CGG codon (in green) in m$^1$G37 deficiency, increasing from 1.0 in control cells to 3.0 in *trmD-deg* cells and in *trmD-KO* (*trm5*–) cells. Notably, CGG was the only Arg codon where pausing was observed upon loss of m$^1$G37, whereas the other Arg codons CG[A/C/U], AGA, and AGG were not affected. Additionally, we observed a small increase of the pause score at the Leu codon CUA in m$^1$G37 deficiency at 1.4 relative to the pause score of 1.0 in control samples (*Figure 2B*). Together, these data showed significant pauses at all Pro codons CCN, the Arg CGG codon, and to a lesser extent the Leu CUA codon. All of the paused codons are translated by tRNAs that are substrates of m$^1$G37 methylation, although pauses at Leu codons CU[C/G/U], which are translated by tRNAs that should also be methylated with m$^1$G37, were not observed.

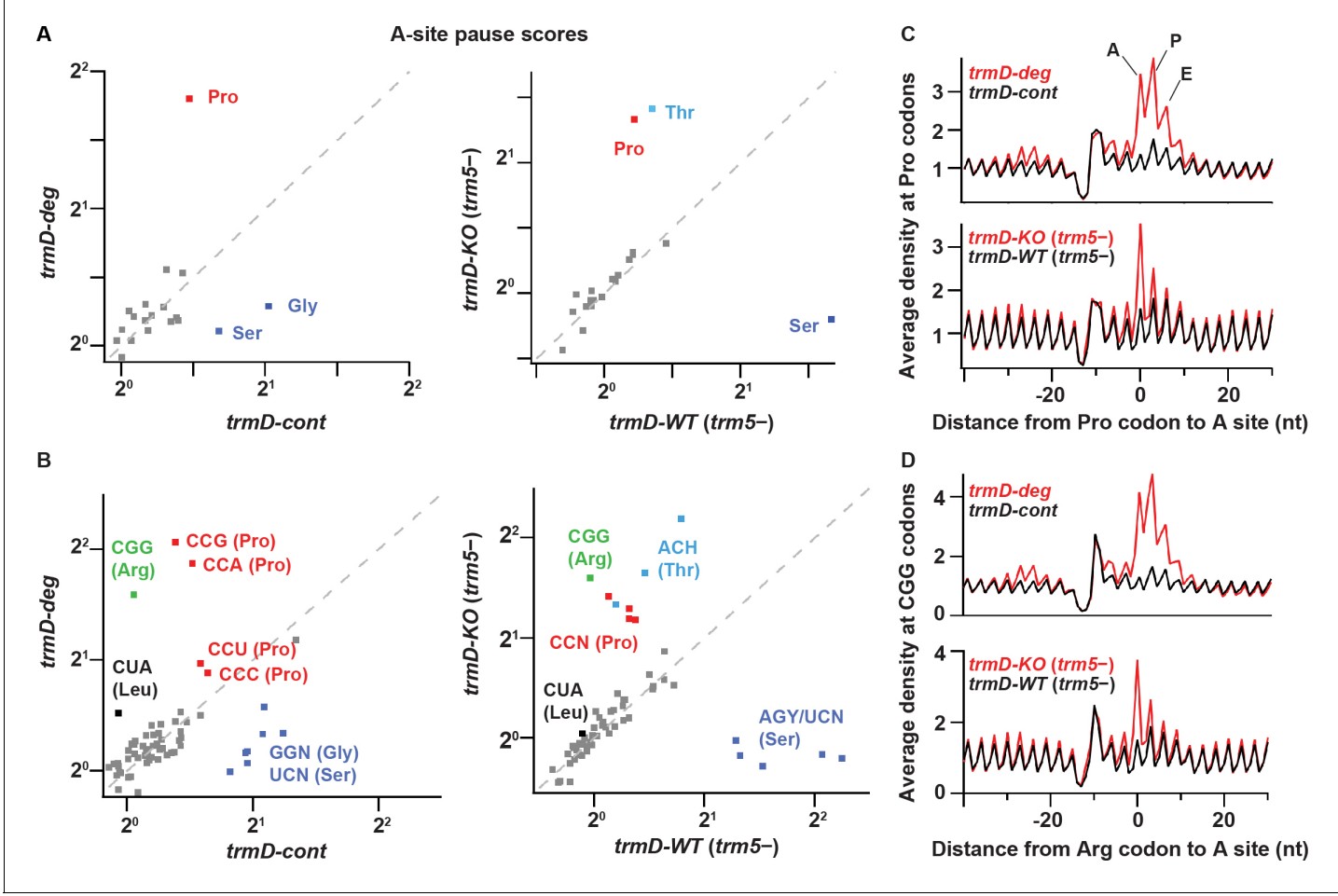

**Figure 2.** TrmD depletion leads to ribosome pausing at CCN (Pro) and CGG (Arg) codons. (A) Pause scores for codons positioned in the ribosomal A site. The codons are grouped by amino acid. Left panel: data from biological replicates of *trmD-deg* and *trmD-cont* strains upon ClpXP induction. Each biological replicate is an independent culture of cells. Right panel: data from *trmD-KO* (*trm5–*) and *trmD-WT* (*trm5–*) strains upon turning off *trm5*. (B) The same as above but showing the 61 sense codons individually. (C) Plots of average ribosome density aligned at Pro codons for *trmD-deg* strain (top) and *trmD-KO* (*trm5–*) strain (bottom) in red with their respective control strains in black. The peaks corresponding to the ribosomal A, P, and E sites are labeled. (D) Plots of average ribosome density aligned at CGG codons as above. Note that *trmD-deg* samples were obtained with the traditional ribosome profiling lysis buffer, whereas *trmD-KO* (*trm5–*) samples were obtained with a high Mg$^{2+}$ lysis buffer that more effectively arrests translation after cell lysis, revealing these pauses at higher resolution.

The increase in the average pause scores in m$^1$G37 deficiency indicates accumulation of ribosome density when the affected codon is positioned in the A site. To explore the possibility of other effects on translation, we looked more broadly at the average ribosome density aligned to a specific codon of interest. For *trmD-deg* vs. *trmD-cont* samples, we observed strong increases of ribosome density in the A, P, and E sites at Pro codons, and at the Arg CGG codons in m$^1$G37 deficiency, with the highest peak in the ribosomal P site (*Figure 2C–D*, top). Although these findings could be interpreted to mean that m$^1$G37 deficiency in tRNAs for these codons led to pausing when each tRNA was in the P or E site, we recognized that the *trmD-deg* and *trmD-cont* samples were obtained using the traditional lysis buffer with chloramphenicol to arrest translation in the lysate. We have since discovered that ribosomes continue to translate a few codons in the lysate under these conditions (*Mohammad et al., 2019*), suggesting that the several peaks observed in m$^1$G37 deficiency likely arose from a strong pause at the A site that was blurred by ongoing translation in the lysate during sample preparation. In contrast, the *trmD-KO* (*trm5–*) and *trmD-WT* (*trm5–*) samples were prepared with a buffer containing 150 mM MgCl$_2$, which immediately arrests ribosomes without the need for antibiotics (*Mohammad et al., 2019*). In these samples, we observed that pauses at Pro codons and

at Arg CGG codons (*Figure 2C–D*, bottom) were the strongest when each codon was positioned at the A site, without an increased density in the P and E sites. We conclude that pauses are primarily in the A site, due to defects in decoding associated with m$^1$G37 deficiency.

Interestingly, we observed increased ribosome density ~25 nt upstream of affected codons in *trmD-deg* relative to *trmD-cont* samples (*Figure 2C–D*, top). This distance is roughly equivalent to the footprint length of a single ribosome, indicating that the increased density was due to collision of an upstream ribosome with a paused ribosome that was struggling to decode an affected codon. The collision of two ribosomes suggests that the pausing of the downstream ribosome at the affected codon is significantly prolonged. These findings are consistent with our prior observation of pausing and formation of disomes upon treatment of cells with mupirocin, an antibiotic that blocks aminoacylation of tRNA$^{Ile}$ (*Mohammad et al., 2019*).

## Reduced aminoacylation and A-site peptide-bond formation of m$^1$G37-deficient tRNAs

The observed ribosome pausing at specific codons in m$^1$G37 deficiency raised two possibilities for the tRNAs translating these codons. First, m$^1$G37 deficiency may reduce aminoacylation of these tRNAs by the respective aminoacyl-tRNA synthetases (aaRSs), preventing them from forming a ternary complex (TC) with EF-Tu-GTP and entering the ribosomal A site. Second, m$^1$G37 deficiency may prevent these tRNAs from peptide-bond formation at the A site, leading to ribosome stalling. We addressed these two possibilities, which are not mutually exclusive, with all three isoacceptors of tRNA$^{Pro}$ and the tRNA$^{Arg}$(CCG) isoacceptor. To isolate the effect of m$^1$G37, we prepared tRNAs as T7 RNA polymerase (RNAP) transcripts lacking m$^1$G37 or any other post-transcriptional modification (the G37-state) and compared their activity to transcripts that were subsequently modified with m$^1$G37 by TrmD *in vitro* (the m$^1$G37-state). We confirmed that the level of methylation in the m$^1$G37-state was high, reaching nearly 100% for the m$^1$G37-state of *E. coli* tRNA$^{Pro}$(UGG) and nearly 70% for the m$^1$G37-state of *E. coli* tRNA$^{Arg}$(CCG) (*Figure 3—figure supplement 1A*). In addition, we also purified some of these tRNAs from cells (the native-state) containing the full complement of natural post-transcriptional modifications.

Aminoacylation of tRNA$^{Pro}$ with Pro was performed with purified *E. coli* ProRS under steady-state multi-turnover conditions (*Zhang et al., 2006*). The initial rate as a function of the concentration of each tRNA was measured and the data were fit to the Michaelis-Menten equation to derive kinetic parameters $K_m$ (tRNA) and $k_{cat}$. For all three isoacceptors of tRNA$^{Pro}$, the catalytic efficiency $k_{cat}/K_m$ was decreased from the m$^1$G37-state to the G37-state by 3- to 10-fold (*Figure 3A* and Figure 3—source data 1). The reduction in $k_{cat}/K_m$ was driven by an increase in $K_m$ for all three tRNAs, indicating that m$^1$G37 may be important for binding of each tRNA to ProRS (Figure 3—source data 1A). This apparent binding defect makes sense structurally because m$^1$G37 is immediately downstream of the two conserved anticodon nucleotides G35-G36, which are the major determinants of tRNA$^{Pro}$ binding to ProRS (*Cusack et al., 1998*; *Yaremchuk et al., 2000*; *Yaremchuk et al., 2001*). The largest decrease in $k_{cat}/K_m$ (10-fold) upon loss of m$^1$G37 was observed for the UGG isoacceptor. Unique among the isoacceptors of tRNA$^{Pro}$, the UGG isoacceptor is required for cell growth and survival (*Nasvall et al., 2004*) and is most critically dependent on m$^1$G37 for maintaining the translational reading-frame (*Gamper et al., 2015a*). The critical role that m$^1$G37 plays in the UGG isoacceptor was further highlighted by the finding that aminoacylation reaction with the m$^1$G37-state tRNA has the same $k_{cat}/K_m$ values as the native-state tRNA (Figure 3—source data 1A), indicating that all other post-transcriptional modifications played little or no role in aminoacylation.

Aminoacylation of the tRNA$^{Arg}$(CCG) isoacceptor with Arg was performed with purified *E. coli* ArgRS under steady-state multi-turnover conditions. The $k_{cat}/K_m$ of aminoacylation was decreased from the m$^1$G37-state to the G37-sate by 5.6-fold (*Figure 3A* and Figure 3—source data 1A). This decrease was largely driven by a loss in $k_{cat}$ (Figure 3—source data 1A), indicating that m$^1$G37 contributed to catalysis, which is a different effect than that observed for tRNA$^{Pro}$, where loss of m$^1$G37 likely reduced tRNA binding to ProRS. Additionally, we observed a loss of 14.3-fold in $k_{cat}/K_m$ from the native-state to the G37-state (Figure 3—source data 1A), greater than the 5.6-fold loss due to m$^1$G37 alone, indicating that other post-transcriptional modifications played a role in aminoacylation of this tRNA$^{Arg}$, which also contrasts the observation that m$^1$G37 alone is sufficient for rapid aminoacylation of tRNA$^{Pro}$. Together, these results show that m$^1$G37 is required for efficient aminoacylation

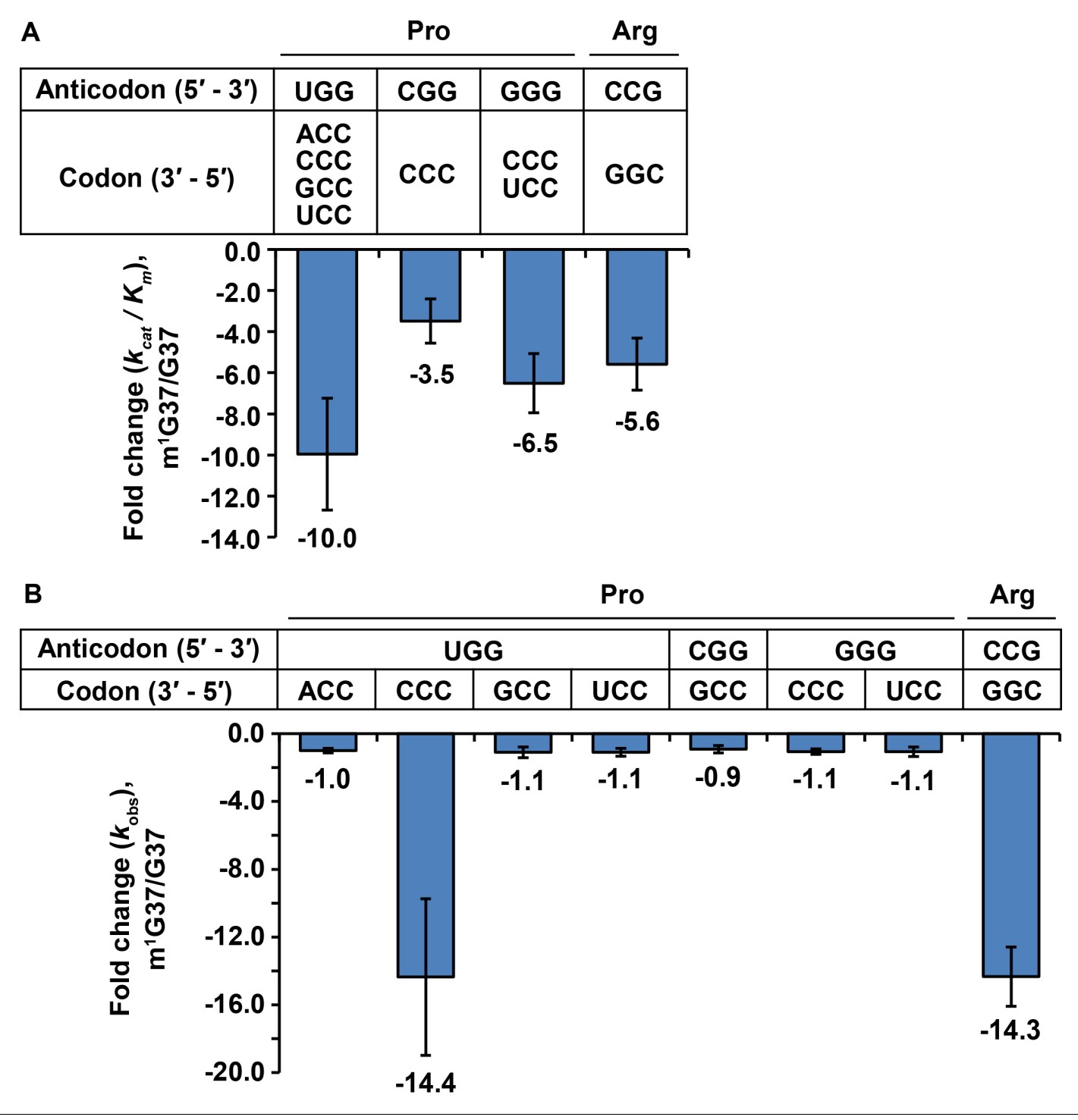

**Figure 3.** Kinetic parameters showing the loss of tRNA aminoacylation and peptide-bond formation at the A site in m$^1$G37 deficiency. (**A**) Fold-change in the loss of $k_{cat}/K_m$ of tRNA aminoacylation from the m$^1$G37-state to the G37-state. Bar graphs are shown for the UGG, the CGG, and the GGG isoacceptors of *E. coli* tRNA$^{Pro}$, and for the CCG isoacceptor of *E. coli* tRNA$^{Arg}$, with the fold-change value displayed at the bottom of each bar. The bars are SD of three independent replicates (n = 3), and the data are presented as the mean ± SD for each sample. (**B**) Fold-change in the loss of $k_{obs}$ from the m$^1$G37-state to the G37-state in overall reaction at the A site leading to peptide-bond formation. Bar graphs are shown for the UGG, CGG, and GGG isoacceptors of *E. coli* tRNA$^{Pro}$, and for the CCG isoacceptor of *E. coli* tRNA$^{Arg}$. Each tRNA is paired to the cognate codon, and the fold-change value is displayed at the bottom of each bar as the mean ± SD of three independent replicates (n = 3).

The online version of this article includes the following figure supplement(s) for figure 3:

*Figure 3 continued on next page*

Figure 3 continued

**Figure supplement 1.** The in vitro reactions for tRNA methylation and peptide-bond formation.

of all three isoacceptors of tRNA$^{Pro}$ and the tRNA$^{Arg}$(CCG) isoacceptor, and that loss of m$^1$G37 reduces aminoacylation in all cases, although by apparently different mechanisms.

To determine whether m$^1$G37 deficiency reduced peptide-bond formation with affected tRNA in the A site, we used our *E. coli in vitro* translation system of purified components and supplemented it with requisite tRNAs and translation factors to perform a series of ensemble rapid kinetic studies (*Gamper et al., 2021*; *Gamper et al., 2015a*; *Gamper et al., 2015b*). We assayed for the synthesis of a peptide bond between the [$^{35}$S]-fMet moiety of the P site [$^{35}$S]-fMet-tRNA$^{fMet}$ of a 70S initiation complex (70S IC) and the aminoacyl moiety of a TC delivered to the A site. This assay monitored all of the reactions at the A site, including the initial binding of the TC to the A site, EF-Tu-catalyzed GTP hydrolysis, accommodation of the aa-tRNA to the A site, and peptidyl transfer. Each TC carried a fully aminoacylated tRNA in the G37-state or in the m$^1$G37-state, in excess of the 70S IC, to allow evaluation of the activity of peptide-bond formation. The TC was kept at a limiting concentration (0.2 μM), relative to the previously reported $K_d$ (3–4 μM) (*Cochella and Green, 2005*), such that the dipeptide-bond formation catalyzed by a limiting 70S IC (0.1 μM) proceeded linearly over time, as shown for tRNA$^{Pro}$(UGG) against the CCA codon (*Figure 3—figure supplements 1B and C*) and for tRNA$^{Arg}$(CCG) against the CGG codon (*Figure 3—figure supplement 1D and E*). The slope of each linear production ($k_{obs}$) under these limiting conditions represented the catalytic efficiency $k_{cat}/K_m$ of the overall reaction of peptide-bond formation at the A site, which is a composite term that could be limited by any of the intermediate steps.

For tRNA$^{Pro}$, the UGG isoacceptor was assayed at all four Pro codons CCN, the CGG isoacceptor was assayed at the codon CCG, and the GGG isoacceptor was assayed at the codons CC[C/U] (*Figure 3B* and Figure 3—source data 1B). The results of these assays showed that, of all of the tested anticodon-codon pairs, loss of m$^1$G37 only had a significant effect on $k_{obs}$ with the UGG iso-acceptor at the CCC codon, decreasing $k_{obs}$ by 14.4-fold from the m$^1$G37-state to the G37-state (*Figure 3B* and Figure 3—source data 1B). Thus, in contrast to aminoacylation, where all isoaccep-tors of tRNA$^{Pro}$ were affected by loss of m$^1$G37, loss of m$^1$G37 only affected the overall reaction of dipeptide formation for one isoacceptor at one codon. Likewise, the CCG isoacceptor of tRNA$^{Arg}$ was assayed at the cognate codon CGG, showing a significant decrease of $k_{obs}$ by 14.3-fold from the m$^1$G37-state to the G37-state (*Figure 3B* and Figure 3—source data 1B). Thus, for the single iso-acceptor of tRNA$^{Arg}$, loss of m$^1$G37 reduced the activity of both aminoacylation and the overall reac-tion of dipeptide formation at the A site.

## Loss of aminoacylation in m$^1$G37-deficient cells

Following up on the results of kinetic studies *in vitro*, we investigated whether m$^1$G37 deficiency led to loss of aminoacylation of affected tRNAs *in vivo*. Total RNA was extracted from *E. coli trmD-KO* (*trm5+*) and *trmD-KO* (*trm5–*) cells using an acid buffer (pH 4.5) and run on an acid-urea gel to pre-serve the levels of charged aa-tRNA vs. uncharged tRNA at the time of harvest. Northern blots with probes against tRNA$^{Pro}$(UGG) and tRNA$^{Arg}$(CCG) showed that both had reduced aa-tRNA levels, decreasing from 84% to 68% and from 74% to 62% respectively in m$^1$G37 deficiency (*Figure 4A*, left and middle panels). We confirmed that the loss of aminoacylation of these two tRNAs was specific to m$^1$G37 deficiency. Acid-urea gel analysis of *E. coli* tRNA$^{Tyr}$(QUA) (Q: queuosine), which contains ms$^2$i$^6$A37 (2-methyl-thio-$N^6$-isopentenyl adenosine) and is not methylated by TrmD, showed that its aa-tRNA level was similar (78% vs. 81%) whether m$^1$G37 was abundant or deficient (*Figure 4A*, right panel).

The loss of aminoacylation of tRNA$^{Pro}$(UGG) and tRNA$^{Arg}$(CCG) is not due to the loss of the cor-responding charging enzymes encoded by *proS* and *argS*, respectively. RNA-seq and ribosome pro-filing data showed that the expression level of these two enzymes was unaffected by m$^1$G37 deficiency (*Figure 4B*). Moreover, cell lysates in m$^1$G37-abundant (*trm5+*) and m$^1$G37-deficient (*trm5–*) conditions exhibited a similar aminoacylation activity when assayed with the methylated m$^1$G37-tRNA$^{Pro}$(UGG) or m$^1$G37-tRNA$^{Arg}$(CCG) (*Figure 4C*, left and middle panels), indicating that the enzymatic activity of *proS* and *argS* was similar between the two growth conditions. As

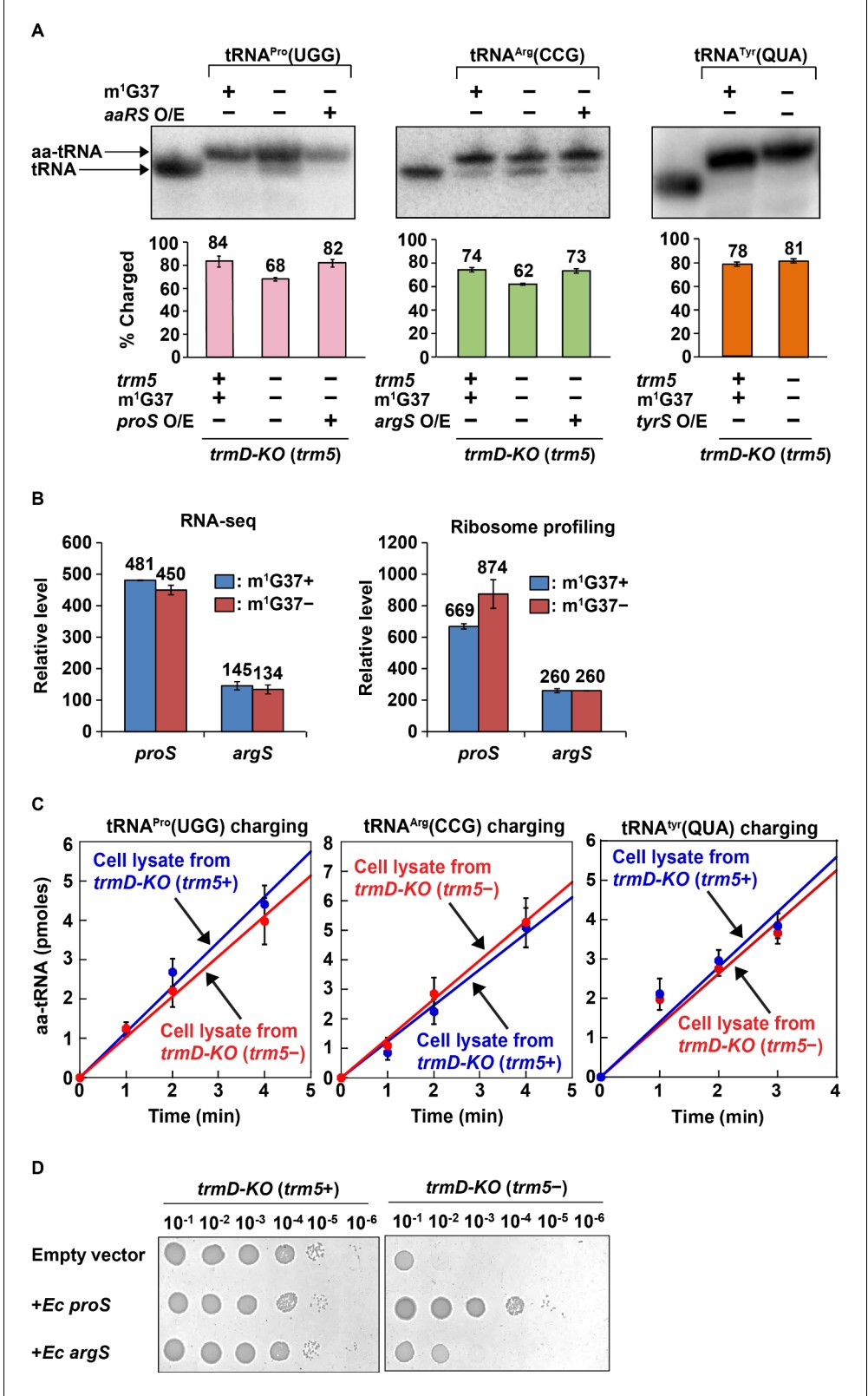

**Figure 4.** Acid-urea gel analysis of loss of tRNA aminoacylation in m$^1$G37-deficient *E. coli* cells. (**A**) Northern blots of acid-urea gels showing the fractional distribution of charged (aa-tRNA) vs. uncharged (tRNA) in total RNA prepared from *trmD-KO* cells complemented by the Ara-controlled *trm5*. Three *E. coli* tRNAs were probed: tRNA$^{Pro}$(UGG) (left panel), tRNA$^{Arg}$(CCG) (middle panel), and tRNA$^{Tyr}$(QUA) (right panel), each under m$^1$G37-abundant (+) and m$^1$G37-deficient (–) conditions. Of the three, the m$^1$G37-dependent tRNA$^{Pro}$(UGG) and tRNA$^{Arg}$(CCG) were further probed in m$^1$G37-

*Figure 4 continued on next page*

*Figure 4 continued*

deficient cells with over-expression (O/E) of *proS* and *argS*, respectively. In contrast, the m$^1$G37-independent tRNA$^{Tyr}$(QUA) served as a control and was not probed for O/E of *tyrS*. The % of aa-tRNA in each case is calculated from the sum of the charged aminoacyl-tRNA (aa-tRNA) and the uncharged tRNA and is shown in the bar graph below for three biological replicates (n = 3). (B) Relative levels of expression of *E. coli proS* and *argS* in RNA-seq and ribosome-profiling analysis in m$^1$G37+ and m$^1$G37– conditions of two biological replicates. (C) Aminoacylation activity in *E. coli* cell lysates prepared in m$^1$G37+ and m$^1$G37– conditions. Three enzymes, encoded by *proS* (left panel), *argS* (middle panel), and *tryS* (right panel), were each tested against the cognate tRNA (i.e., m$^1$G37-state tRNA$^{Pro}$(UGG), m$^1$G37-state tRNA$^{Arg}$(CCG), and A37-state tRNA$^{Tyr}$(QUA), respectively). Data are the average of three biological replicates (n = 3). Aminoacylation was performed under substrate-limiting conditions to obtain a linear line of product formation over time. (D) Viability of *E. coli trmD-KO* (*trm5+*) and *trmD-KO* (*trm5–*) cells harboring an empty vector, or the vector over-expressing *E. coli proS* or *argS*, was evaluated by spotting a serial dilution of cells onto an M9 plate with or without Ara for creating *trmD-KO* (*trm5+*) and *trmD-KO* (*trm5–*) conditions.

expected, aminoacylation of the control tRNA$^{Tyr}$(QUA) was also similar between the two cell lysates (*Figure 4C*, right panel), validating that equivalent amounts of cell lysates were added to the different reactions.

Additional data showed that the lost aminoacylation of tRNA$^{Pro}$(UGG) and tRNA$^{Arg}$(CGG) in m$^1$G37 deficiency was recovered by over-expression of *proS* and *argS* in *trmD-KO* (*trm5–*) cells (*Figure 4A*, left and middle panels, the third lane of each). This recovery indicates that the poor aminoacylation kinetics of m$^1$G37-deficient tRNA can be overcome by increasing the corresponding aaRS enzyme. We asked whether these higher levels of aminoacylation would restore cell viability by improving the translation activity of m$^1$G37-deficient tRNAs. Indeed, while control cells in m$^1$G37-abundant conditions (*trmD-KO* (*trm5+*)) grew robustly, with or without over-expression of a plasmid-borne *proS* or *argS*, m$^1$G37-deficient cells (*trmD-KO* (*trm5–*)) grew poorly with an empty plasmid, but improved viability upon over-expression of *proS* or *argS* (*Figure 4D*). Notably, while the loss of aminoacylation in m$^1$G37 deficiency was similar to tRNA$^{Pro}$(UGG) and tRNA$^{Arg}$(CCG) in cell lysates (*Figure 4A*, left and middle panels), the recovery of cell viability upon over-expression of *proS* approached that of the positive control (*trmD-KO* (*trm5+*)), whereas that of over-expression of *argS* was much weaker (*Figure 4D*). This indicates that it is the loss of aminoacylation of tRNA$^{Pro}$ species that is the driver for limiting cell growth under m$^1$G37 deficiency.

Taken together, these findings show that m$^1$G37 deficiency reduces the aminoacylation level of all tRNA$^{Pro}$ species and tRNA$^{Arg}$(CCG) *in vitro* and *in vivo*. This loss of aminoacylation levels compromises cell viability but can be overcome by over-expression of *proS* and to a lesser extent over-expression of *argS*.

## Changes in gene expression by m$^1$G37 deficiency

We next asked what changes in gene expression took place in m$^1$G37 deficiency. Using the DESeq algorithm (*Love et al., 2014*) to analyze the RNA-seq data, we identified genes whose steady-state levels of mRNAs were altered with statistical significance by loss of m$^1$G37. We found 220 genes with more than 2-fold higher expression (p < 0.01) in the *trmD-deg* strain (shown in red, *Figure 5A*). Conversely, we identified 166 genes that were repressed by more than 2-fold (p < 0.01, colored in blue, *Figure 5A*). For both the up- and down-regulated genes, we identified several pathways that are affected and whose genes are enriched at statistically significant levels (*Figure 5C*). We also observed changes at the translational level. Using the Xtail algorithm (*Xiao et al., 2016*) to identify changes in ribosome occupancy (RO = Ribo-seq density/RNA-seq density), we found 71 genes with higher levels of apparent ribosome occupancy (1.7- to 6.5-fold increase) in the *trmD-deg* strain relative to the *trmD-cont* strain (p < 0.01, *Figure 5B*). Because the magnitude of transcriptional changes was greater than that of translational changes, yielding clearer results on which pathways are affected, we focused on the transcriptional changes arising from m$^1$G37 deficiency.

Many of the genes with higher RNA levels upon TrmD depletion are involved in amino acid biosynthesis and transport (*Figure 5C*). For example, induction of the Leu operon (*leuA*, *leuB*, *leuC*, and *leuD*) was particularly strong (~16-fold). This Leu operon is regulated by transcriptional attenuation depending on the efficiency of translation of the upstream *leuL* leader sequence (*Wessler and Calvo, 1981*; *Wohlgemuth et al., 2013*). As with leader sequences of other amino acid biosynthesis operons (*Kolter and Yanofsky, 1982*), translation of *leuL* serves as a sensitive genetic switch that has evolved to sense levels of aa-tRNAs and to respond to amino acid starvation by up-regulating

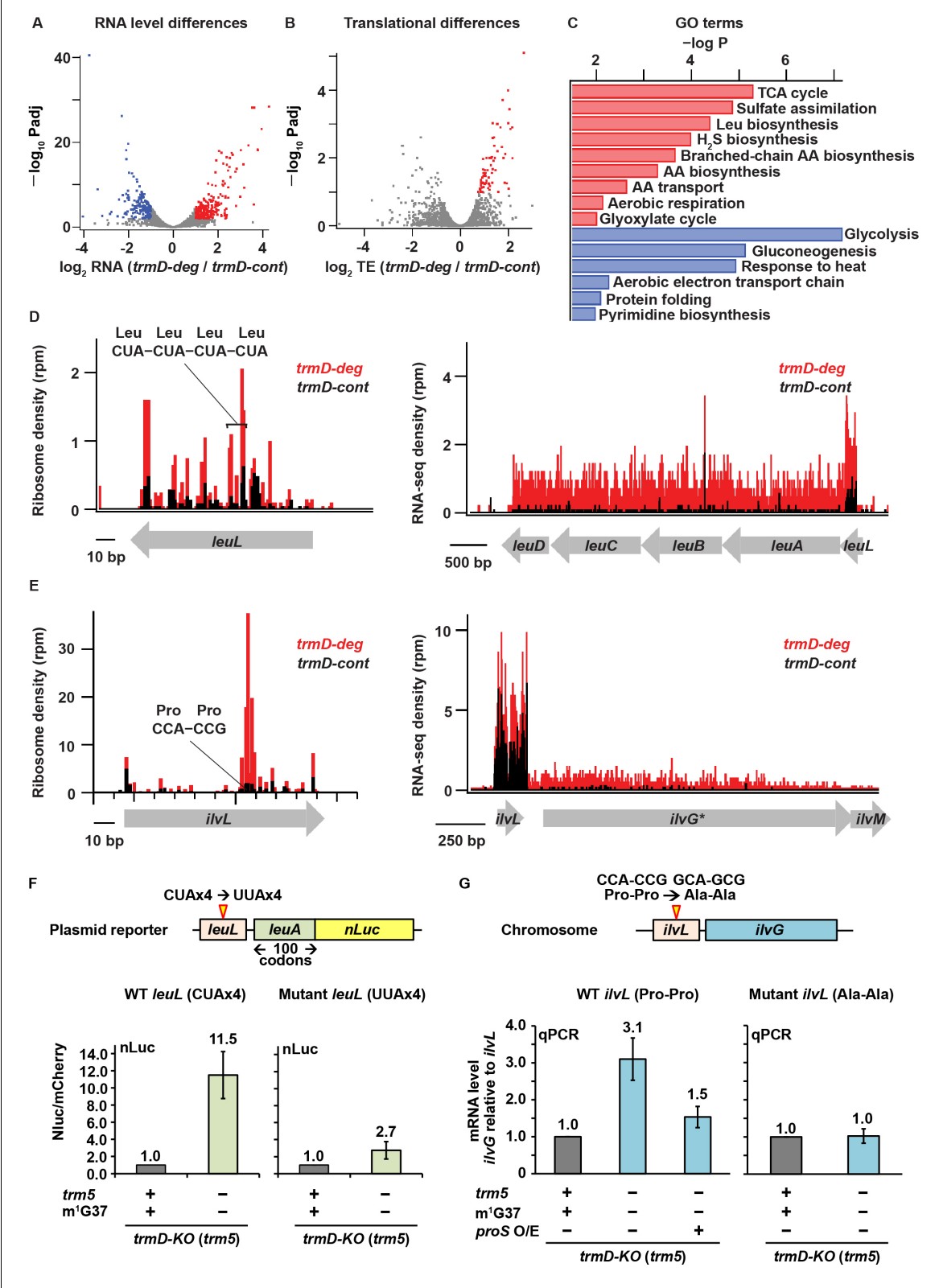

**Figure 5.** Changes in gene expression arising from m$^1$G37 deficiency. (**A**) Volcano plot showing differences in steady-state RNA levels (from RNA-seq data analyzed by DESeq) using samples of two replicates of *trmD-deg* and *trmD-cont* strains upon ClpXP induction. Genes that are more than 2-fold higher in the mutant with p < 0.01 are colored red (n = 220); those that are more than 2-fold lower with p < 0.01 are colored blue (n = 166). (**B**) Volcano plot showing differences in ribosome occupancy (Ribo-seq/RNA-seq analyzed by Xtail) using samples from two replicates of *trmD-deg* and *trmD-cont*

*Figure 5 continued on next page*

*Figure 5 continued*

strains upon ClpXP induction. Of these, 95 genes that are up-regulated (p < 0.1) are colored red. (C) Gene ontology categories for genes enriched in the up-regulated genes (red) and down-regulated genes (blue) from (A). (D) Gene model showing ribosome reads (in rpm) and RNA-seq density (in rpm, reads per million mapped reads) at the *leuL* leader sequence upstream of the *leuABCD* operon. (E) Gene model showing ribosome reads (in rpm) and RNA-seq density (in rpm) at the *ilvL* leader sequence upstream of the *ilvGMEDA* operon. (F) A plasmid reporter construct that demonstrates the codon-specific effect in the *leuL* leader sequence on expression of the downstream *leuA* gene. The plasmid reporter encodes the genomic sequence of *leuL,* followed by the first 100 codons of *leuA*, in the native sequence or with substitution of the four consecutive CUA codons with four UUA codons. The 100 codons of *leuA* are then fused to the nano-luciferase (nLuc) gene in-frame. Expression of nLuc, normalized by co-expression of mCherry in a separate plasmid within the same cell, is shown for $m^1G37+$ and $m^1G37-$ conditions for the average of three biological replicates (n = 3). (G) Analysis of the CCA-CCG (Pro-Pro) codons in the genomic locus of *ilvL* as the determinant of regulation of gene expression of the downstream *ilvG* gene. A variant *Escherichia coli* strain was created that changed the CCA-CCG (Pro-Pro) codons to GCA-GCG (Ala-Ala) codons at the natural genomic locus, using CRISPR/Cas editing. Expression of *ilvG* was monitored by qPCR analysis in $m^1G37+$ and $m^1G37-$ conditions and in $m^1G37-$ condition with over-expression of *proS* as shown as the average of three biological replicates (n = 3).

the downstream biosynthetic pathways (*Wohlgemuth et al., 2013*). We observed ribosomal pausing in *leuL* at the four consecutive Leu CUA codons in $m^1G37$ deficiency (*Figure 5D*), consistent with our observation of ribosome pausing at CUA codons (*Figure 2B*), suggesting that the pausing would prevent transcriptional termination and allow transcription elongation into the downstream Leu bio-synthetic genes, resulting in elevated RNA levels. In agreement with this transcriptional attenuation mechanism, we observed in the control *trmD-cont* strain efficient transcriptional termination down-stream of *leuL*, where the average RNA-seq density (adjusted for length) decreased 29-fold from the *leuL* leader sequence to the first gene in the operon (*leuA*), whereas we observed in the *trmD-deg* strain only a 4-fold decrease (*Figure 5D*). These results support the notion that ribosome pausing at the four consecutive Leu CUA codons in *leuL* during $m^1G37$ deficiency reduced transcriptional termi-nation of the downstream *leuA* gene (*Figure 5D*).

Another example is the *ilvL* leader sequence upstream of the *ilvGMEDA* operon encoding genes for biosynthesis of branched chain amino acids Ile, Leu, and Val. Ribosome pausing in *ilvL* would pre-vent transcriptional termination and allow expression of the downstream biosynthetic genes (*Nargang et al., 1980*). Although this genetic switch has evolved to sense changes in levels of aa-tRNAs associated with the amino acids produced by the downstream genes, we instead observed strong ribosome pauses at two consecutive Pro codons CCA-CCG in $m^1G37$ deficiency (*Figure 5E*). These strong pauses within the *ilvL* leader sequence could trigger anti-termination, explaining the higher RNA-seq levels of the downstream gene *ilvG*. Indeed, while we observed in the control strain a 45-fold decrease in the RNA-seq density from the region around *ilvL* to the first half of the down-stream *ilvG* gene, we observed in the *trmD-deg* strain only a 9-fold decrease in the RNA-seq density (*Figure 5E*). Notably, the *ilvG* gene in strains derived from *E. coli* K12 MG1655, such as the one used here, has a mutation that induces a frameshift midway through the gene, suggesting that it may be a pseudogene, although the *ilvG* gene appears intact in other *E. coli* strains.

We confirmed that the changes of expression of these two operons were due to pauses at spe-cific codons induced by $m^1G37$ deficiency in cognate tRNAs. To examine the effect of the *leuL* leader sequence on expression of *leuA*, we generated a plasmid reporter construct containing *leuL* and the first 100 codons of *leuA* fused in-frame to the nano-luciferase (nLuc) gene (*Figure 5F*). Upon expression of the WT reporter construct, $m^1G37$ deficiency in *trmD-KO* (*trm5–*) cells elevated the nLuc readout by 11.5-fold relative to control cells, consistent with the notion that ribosome stalling at the four consecutive Leu CUA codons in *leuL* prevents transcriptional termination and allows tran-scription of the downstream *leuA-nLuc* fusion. In contrast, a smaller activation during $m^1G37$ defi-ciency (only 2.7-fold) was observed with a second nLuc reporter where the four consecutive Leu CUA codons were changed to Leu UUA codons, which do not require $m^1G37$ for translation.

To examine the effect of the *ilvL* leader sequence on expression of *ilvG*, we altered the two Pro codons CCA-CCG in *ilvL* to two Ala codons GCA-GCG by genome editing using CRISPR and fol-lowed the expression of the downstream *ilvG* gene with qPCR (*Figure 5G*). With the wild-type *ilvL* sequence, *ilvG* was up-regulated 3.1-fold in $m^1G37$-deficient *trmD-KO* (*trm5–*) cells relative to $m^1G37$-abundant *trmD-KO* (*trm5+*) cells (left panel, bar 2). In contrast, no increase was observed for the mutant *ilvL* sequence (right panel, bar 2). These results support the notion that ribosome stalling at the Pro codons in *ilvL* activates expression of the downstream *ilvG* in $m^1G37$ deficiency.

Furthermore, given our observation that over-expression of *proS* restored aminoacylation and cell viability in m$^1$G37 deficiency (*Figure 4D*), we asked if it might also minimize pausing and induction of this genetic switch. We found that over-expression of *proS* in *trmD-KO* (*trm5–*) cells with the wild-type *ilvL* leader sequence reduced the activation of *ilvG* from 3.1-fold to 1.5-fold (left panel, bars 2 and 3). This result is consistent with the notion that elevated *proS* expression restores aminoacylation levels of tRNA$^{Pro}$ in cells, even in m$^1$G37 deficiency, relieving ribosome stalling and repressing the downstream *ilvG* gene.

In addition to Leu, Ile, and Val, the biosynthetic pathways for Trp, His, and Cys were highly up-regulated in the *trmD-deg* strain (*Figure 5C*), even though there is no evidence of ribosome stalling at the 5'-leader of the relevant operons. The up-regulation of these pathways suggests that they are activated by a genome-wide response due to ribosome stalling at Pro CCN, Arg CGG, and Leu CUA codons (*Figure 2B*). The increase in expression of the Cys biosynthesis pathway is correlated with the enrichment in gene ontology (GO) terms for genes involved in sulfate assimilation and hydrogen sulfide biosynthesis (*Figure 5C*) and is consistent with the notion that cysteine is synthesized from serine in bacteria by incorporation of sulfide to *O*-acetylserine (*Kredich and Tomkins, 1966*).

## Changes of gene expression in m$^1$G37 deficiency consistent with the stringent response

The high levels of expression of amino acid biosynthesis genes in m$^1$G37 deficiency was reminiscent of the bacterial stringent response, which is triggered in nutrient starvation by uncharged tRNAs binding to the ribosomal A site, activating synthesis of ppGpp catalyzed by the RelA protein (*Gourse et al., 2018*). In the stringent response, ppGpp binds to two sites on RNA polymerase (RNAP) and re-programs the transcriptional landscape (*Gourse et al., 2018*; *Ross et al., 2016*; *Ross et al., 2013*), shutting down gene expression of rRNAs and ribosome proteins, while up-regulating amino acid biosynthesis genes to respond to amino acid starvation. Our observation that m$^1$G37 deficiency reduced aminoacylation of all isoacceptors of tRNA$^{Pro}$ and the tRNA$^{Arg}$(CCG) iso-acceptor raised the possibility that these effects would induce the programmatic changes in gene expression similar to those in the stringent response.

To test this possibility, we compared the changes in RNA levels in m$^1$G37 deficiency with the changes in RNA levels upon induction of the stringent response. We used the dataset for the stringent response obtained by Gourse and co-workers that compared changes in RNA levels between an *E. coli* strain containing the ppGpp-binding sites in RNAP and a variant lacking the binding sites upon over-expression of a *relA* mutant that was constitutively activated to synthesize ppGpp (*Sanchez-Vazquez et al., 2019*). This dataset focused on *E. coli* genes that exhibited transcriptional up- and down-regulation in response to direct binding of ppGpp to RNAP, rather than indirect effects from starvation-induced changes in metabolism. After 5 min of induction of the *relA* mutant, the dataset identified a set of 401 down-regulated genes (by >2-fold), a set of 321 up-regulated genes (by >2-fold), and a set of 3036 genes with less than 2-fold changes (*Sanchez-Vazquez et al., 2019*). Using these same sets of genes, we found in our RNA-seq data that genes that were down-regulated upon *relA* over-expression were also down-regulated in m$^1$G37 deficiency (log$_2$(median) = –0.36; *t*-test p = 8.9 × 10$^{-15}$) (*Figure 6A*). Likewise, genes that were up-regulated in *relA* over-expression were also up-regulated in m$^1$G37 deficiency (*Figure 6A*), although the extent of up-regulation in m$^1$G37 deficiency was smaller (log$_2$(median) = 0.29; *t*-test p = 2.3 × 10$^{-11}$). Additionally, the non-responsive genes showing less than 2-fold changes upon *relA* over-expression also showed little change in m$^1$G37 deficiency (log$_2$(median) ~ 0.04). Together, these results show that genes that are strongly induced or repressed by the stringent response are affected in a similar way by m$^1$G37 deficiency, arguing that m$^1$G37 deficiency induces changes in gene expression similar to those of the stringent response.

Following up on this observation, we next asked to what extent two characteristic pathways known to be regulated by the stringent response were also affected by m$^1$G37 deficiency. In nutrient starvation, bacterial cells down-regulate ribosome biosynthesis due to the reduced demand for protein production, but up-regulate amino acid biosynthesis (*Gourse et al., 2018*). As expected, the published RNA-seq data upon *relA* over-expression showed that 89 genes associated with amino acid biosynthesis in the EcoCyc database were more highly expressed than other genes (log$_2$(median) = 0.59, Mann-Whitney p = 4.8 × 10$^{-12}$). Intriguingly, we found that these genes were also up-regulated to a similar extent in m$^1$G37 deficiency (log$_2$(median) = 0.58, Mann-Whitney p = 1.2 × 10$^−$

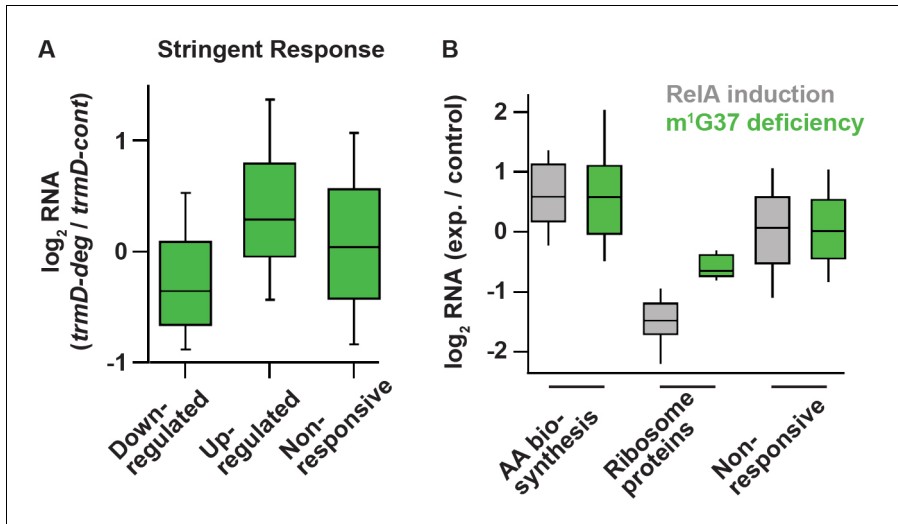

**Figure 6.** Deficiency of m[1]G37 induces gene expression similar to that of the bacterial stringent response. (**A**) A set of genes published previously by Gourse et al. (*Sanchez-Vazquez et al., 2019*), whose steady-state RNA levels changed more than 2-fold upon induction of the stringent response by RelA over-expression (down n = 401 genes, up n = 321 genes, other = 3036 genes). Using these same sets of genes, the ratio of RNA levels in *trmD-deg* and *trmD-cont* strains upon ClpXP induction are shown. (**B**) Changes in expression in genes involved in amino acid biosynthesis (n = 89), genes encoding ribosomal proteins (n = 47), and other genes.

8) (*Figure 6B*), consistent with the enrichment of these genes in our GO annotation (*Figure 5C*). Likewise, 47 ribosome protein genes were down-regulated upon *relA* over-expression ($\log_2$(median) = −1.48, Mann-Whitney p = $1.6 \times 10^{-24}$) and they were also down-regulated in m[1]G37 deficiency, although to a lesser extent ($\log_2$(median) = −0.64, Mann-Whitney p = $1.1 \times 10^{-12}$) (*Figure 6B*). As a control, we also found that the genes that were not induced in these sets (i.e., not ribosome proteins and not involved in amino acid biosynthesis) were less responsive to *relA* over-expression, and that they were also less responsive in m[1]G37 deficiency (*Figure 6B*). Collectively, the parallel changes in steady-state RNA levels between m[1]G37 deficiency and *relA* over-expression were remarkable, suggesting that loss of m[1]G37 triggers a response similar to that of the *relA*-dependent stringent response.

## Metabolic changes

Some of the most striking changes in gene expression in m[1]G37 deficiency occurred in central metabolic pathways: glycolysis, the citric acid (the tricarboxylic acid [TCA]) cycle, and fatty acid oxidation. For example, 10 genes associated with the TCA cycle were significantly up-regulated in the *trmD-deg* strain, making this pathway the most enriched in the GO terms (*Figure 5C*). Genes in the glyoxylate cycle are also up-regulated. Notably, because the glyoxylate cycle bypasses the decarboxylation reactions and instead replenishes TCA cycle intermediates, it is reasonable that these changes may be to build up metabolic intermediates for amino acid biosynthesis. The two-carbon units entering the TCA cycle likely arise primarily from fatty acid oxidation, whose enzymes are highly up-regulated in m[1]G37 deficiency, as opposed to the glycolysis pathway, whose genes are strongly repressed (*Figure 5C*). Thus, it appears that m[1]G37 deficiency shifts metabolism away from consuming Glc toward consuming fatty acids and that cells prioritize building up metabolic intermediates to support amino acid biogenesis.

## Discussion

Here, we provide a genome-wide view of the effect of m[1]G37 deficiency on protein synthesis upon depletion of TrmD in *E. coli*. This genome-wide view is important, because TrmD is ranked as a high-priority antibacterial target (*White and Kell, 2004*), due to its fundamental distinction from Trm5, and its conservation throughout the bacterial domain, essentiality for bacterial life, and possession

of a small molecule-binding site for drug targeting. The small molecule-binding site in TrmD is unique, consisting of a protein topological knot-fold that holds the methyl donor S-adenosyl methionine in an unusual shape (*Ahn et al., 2003*; *Elkins et al., 2003*; *Ito et al., 2015*), which is different from that in Trm5 and in most other methyl transferases (*Goto-Ito et al., 2009*), indicating the possibility to explore novel chemical space and diversity of drugs targeting TrmD. Additionally, we have shown that TrmD deficiency can disrupt the double-membrane cell-envelope structure of Gram-negative bacteria, thus facilitating permeability of multiple drugs into cells, preventing efflux of these drugs, and accelerating bactericidal action (*Masuda et al., 2019*). This multitude of advantages of targeting TrmD emphasizes that a better understanding of the genome-wide function of its $m^1G37$ product will help develop a successful antibacterial strategy.

Using ribosome profiling, we show that $m^1G37$ deficiency causes ribosome stalling at codons specific for tRNAs that are methylated by TrmD, including strong pauses at all Pro CCN codons and the Arg CGG codon and weak pauses at the Leu CUA codon. Stalling is primarily observed when the affected codons are in the ribosomal A site, indicating a distinct mechanistic feature than that of +1 frameshifting, which occurs in $m^1G37$ deficiency after decoding at the A site and during tRNA translocation to the P site and occupancy within the P site (*Gamper et al., 2021*; *Gamper et al., 2015a*). The importance of $m^1G37$ at the ribosomal A site thus expands the spectrum of its biology, which can be summarized in a model that explains its indispensability throughout the entire elongation cycle of protein synthesis (*Figure 7*). In this model, $m^1G37$ deficiency induces ribosome stalling in the A site, triggering changes in gene expression and possibly affecting mRNA stability and co-translational protein folding. Ribosome stalling can be prolonged if a subset of the affected tRNAs is unable to perform peptide-bond formation in the A site, leading to ribosome collisions that increase the frequency of frameshifting (*Smith et al., 2019*). Even if some of the $m^1G37$-deficient tRNAs manage to enter the A site and participate in peptide-bond formation, they would induce +1 frameshifting during translocation to the P site and destabilize interaction with the ribosomal P site (*Hoffer et al., 2020*), leading to premature fall-off from the ribosome and termination of protein synthesis. In contrast to starvation that can be alleviated by re-supply of nutrients, this series of defects associated with $m^1G37$ deficiency cannot be readily resolved, thus underscoring the essentiality of $m^1G37$ in protein synthesis that is coupled to cell growth and survival.

An important finding of this work is that ribosome stalling in $m^1G37$ deficiency is primarily driven by loss of aminoacylation of affected tRNAs, which is shown in enzyme-based kinetic assays for all isoacceptors examined, and in cell-based acid-urea assays for $tRNA^{Pro}(UGG)$ and $tRNA^{Arg}(CCG)$, two major TrmD-dependent species in bacteria. While the loss of aminoacylation in $m^1G37$ deficiency is not due to the loss of ProRS or ArgRS in cells, it can be restored by over-expression of either enzyme, which improves cell viability. These results clearly demonstrate that the loss of aminoacylation in $m^1G37$ deficiency, and consequently the accumulation of uncharged tRNAs, is the underlying basis of ribosome stalling that induces significant changes in gene expression. Direct changes include up-regulation of the *leuABCD* and *ilvGMEDA* operons. Indirect changes include hundreds of genes that are activated and similar to those in the bacterial stringent response – the central bacterial response to nutrient starvation upon RelA sensing of uncharged tRNAs binding to the A site of translating ribosomes. These changes certainly make sense given that we see higher levels of deacylated tRNAs accumulating in $m^1G37$ deficiency and that the binding of deacylated tRNA to the A site is known to induce (p)ppGpp synthesis by RelA on the ribosome. We did not test the role of RelA in the indirect changes in gene expression in $m^1G37$ deficiency, because the *trmD-deg* and *trmD-KO* (*trm5–*) cells are already substantially compromised in cell viability and the deletion of *relA* is expected to further reduce viability. Another confounding factor is that *E. coli* contains two genes responsible for ppGpp synthesis: *relA* and *spoT*, which have overlapping functions. Upon *relA* deletion, the protein product of *spoT* can still synthesize ppGpp in response to some of the metabolic changes that are observed here. Nonetheless, the changes in gene expression in *trmD* deficiency largely parallel with those observed during the stringent response, consistent with the notion that the loss of aminoacylation in $m^1G37$ deficiency activates the stringent response.

The significance of this work is that it demonstrates a stress response in bacteria similar to the stress response in eukaryotes that is activated by changes of the post-transcriptional modification state of tRNA. In yeast, loss of the 5-methoxycarbonylmethyl ($mcm^5$) and/or the 2-thio ($s^2$) groups from the normal $mcm^5s^2U34$-state tRNA activates the *GCN4*-mediated pathway (*Zinshteyn and Gilbert, 2013*). This *GCN4* pathway in yeast is translationally regulated in response to a variety of

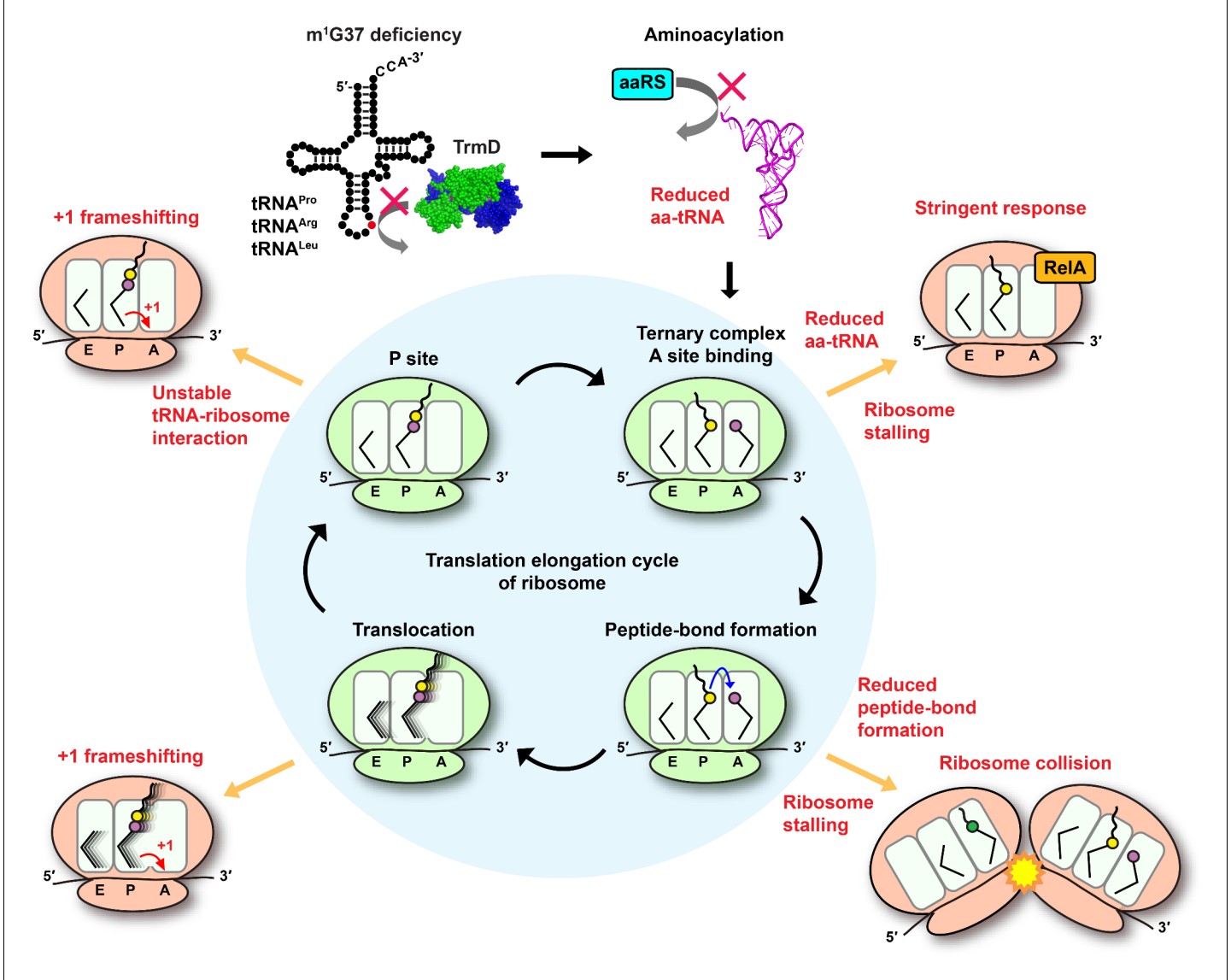

**Figure 7.** Deficiency of m$^1$G37 impairs the elongation cycle of protein synthesis in bacteria. A cloverleaf structure of tRNA, showing the position of m$^1$G37 in red. Loss of m$^1$G37 reduces aminoacylation, lowering levels of aa-tRNAs at the ribosome A site, stalling the ribosome, and activating programmatic changes in gene expression similar to those induced by the bacterial stringent response upon RelA sensing of uncharged tRNA and synthesis of ppGpp. Ribosome stalling at the A site is prolonged if the m$^1$G37-deficient aa-tRNA is delayed in peptide-bond formation, leading to ribosome collision. Even if the m$^1$G37-deficient tRNA manages to enter the A site and participates in peptide-bond formation, it would promote +1 frameshifting during translocation from the A site to the P site and during occupancy within the P site, while also destabilizing the P site structure and possibly falling off from the P site to prematurely terminate protein synthesis. Accumulation of these defects ultimately collapses the entire elongation cycle and leads to cell death.

insults, most notably to amino acid starvation (*Hinnebusch, 2005*). In yeast, loss of the threonyl-carbamoyl group from the normal t$^6$A37-state tRNA also activates the *GCN4* response (*Thiaville et al., 2016*). In *Drosophila*, deficiency of the t$^6$A37-state in a specific tRNA activates the target of rapamycin response (*Rojas-Benitez et al., 2015*), which reprograms gene expression in response to amino acid deficiency. In *Neurospora*, loss of the inosine (I) group from the normal I34-state tRNA activates the CPC-1 (cross-pathway-control-1) response, which is the ortholog of the yeast *GCN4* response (*Lyu et al., 2020*). Mechanistically, loss of the s$^2$ group from the mcm$^5$s$^2$U34-state has been shown to reduce binding and accommodation of tRNA to the ribosomal A site and translocation to the P site (*Ranjan and Rodnina, 2017*). While we had previously shown that loss of the s$^2$ group from the related cmnm$^5$s$^2$U34 (5-carboxy-amino-methyl-2-thio-uridine 34)-state in *E. coli* tRNA$^{Gln}$ also

reduced tRNA aminoacylation and the subsequent binding and accommodation to the A site (*Rodriguez-Hernandez et al., 2013*), we did not investigate the cellular response due to the deficiency in bacteria. Thus, this work on m$^1$G37 deficiency provides a previously unrecognized parallel between *E. coli* and eukaryotes in a stress response that activates amino acid biosynthesis upon loss of a post-transcriptional modification of tRNA. The parallel suggests the evolution of a conserved cellular priority that activates amino acid biosynthesis in response to nutrient deficiency or to tRNA modification deficiency, both of which compromise active protein synthesis and threaten cell viability.

The molecular insight obtained from this study is important for antibacterial therapeutics. The observation that over-expression of *proS* most effectively rescues cell viability in m$^1$G37 deficiency (*Figure 4D*) indicates that aminoacylation of tRNA$^{Pro}$ is a major determinant of cell viability. Notably, loss of aminoacylation of tRNA$^{Pro}$ is uniformly manifested at the $K_m$ (tRNA) step in m$^1$G37 deficiency (Figure 3—source data 1A), suggesting that the immediate proximity of the methylation to the aminoacylation-determinant anticodon G36 nucleotide is important for the bacterial ProRS. In contrast, human ProRS has diverged in tRNA recognition by shifting the emphasis toward the G35 nucleotide, which is more distant from m$^1$G37 (*Burke et al., 2001*). This shift suggests the possibility that the human enzyme may better accommodate m$^1$G37-deficient tRNAs for aminoacylation, and that the eukaryotic response to m$^1$G37 deficiency is executed in a step other than aminoacylation for decoding at the A site. These possibilities remain to be tested in future experiments.

In summary, we show here that m$^1$G37 is required for the ribosomal activity at the A site and that, when combined with previous studies, we emphasize the importance of m$^1$G37 for the entire elongation cycle of protein synthesis. This new insight suggests the possibility that drug targeting of TrmD can be used in combination with classic antibiotics that target the elongation cycle of protein synthesis. For example, targeting TrmD can be explored in combination of antibiotics that target the A site (e.g., tetracycline [Tet], negamycin), those that target the translocation reaction (e.g., viomycin, spectinomycin), and those that target the P site (e.g., streptogramin A, bactobolin A, and retapamulin) (*Vázquez-Laslop and Mankin, 2018*). This combinatorial approach may produce a synergistic and accelerated bactericidal effect that can combat the rapid emergence of multi-drug resistance of pathogens.

# Materials and methods

**Key resources table**

| Reagent type (species) or resource | Designation | Source or reference | Identifiers | Additional information |
|---|---|---|---|---|
| Strain, strain background (*Escherichia coli*) | G78 *trmD-deg* | This paper | | TrmD is fused with a C-terminal YALAA degron tag |
| Strain, strain background (*Escherichia coli*) | G78 *trmD-cont* | This paper | | Control strain without a degron tag |
| Strain, strain background (*Escherichia coli*) | MG1655 *trmD-KO* | *Masuda et al., 2019* | | |
| Strain, strain background (*Escherichia coli*) | MG1655 *trmD-KO ilvL* Ala-Ala mutant | This paper | | CRISPR mutant possessing GCA-GCG codons replacing CCA-CCG codons on *ilvL* gene |
| Strain, strain background (*Escherichia coli*) | JM109 *trmD-KO* | *Demo et al., 2021* | | |
| Strain, strain background (*Escherichia coli*) | BL21(DE3) *trmD-KO* | *Gamper et al., 2015a* | | |
| Strain, strain background (*Escherichia coli*) | BL21(DE3) | CGSC | CGSC#: 12504 | |

*Continued on next page*

*Continued*

| Reagent type (species) or resource | Designation | Source or reference | Identifiers | Additional information |
|---|---|---|---|---|
| Antibody | anti-TrmD (Rabbit polyclonal) | *Li and Björk, 1999* doi: 10.1017/S1355838299980834 | | WB (1:10000) |
| Antibody | anti-rabbit IgG (Goat polyclonal) | Sigma-Aldrich | Cat#: A0545 | WB(1:160000) |
| Peptide, recombinant protein | Proteinase K | Promega | Cat#: V302B | |
| Peptide, recombinant protein | Micrococcal nuclease | Roche | Cat#: 10107921001 | |
| Peptide, recombinant protein | Superscript III reverse transcriptase | Invitrogen | Cat#: 18080–044 | |
| Peptide, recombinant protein | RQ1 RNase-free DNase | Promega | Cat#: M6101 | |
| Commercial assay or kit | TruSeq Stranded Total RNA kit | Illumina | Cat#: 20020598 | |
| Commercial assay or kit | SuperSignal West Pico Chemiluminescent Substrate | Thermo Fisher Scientific | Cat#: 34080 | |
| Commercial assay or kit | NEBuilder HiFi DNA Assembly Cloning Kit | New England Biolabs | Cat#: E5520S | |
| Commercial assay or kit | Nano-Glo Luciferase Assay System | Promega | Cat#: N1110 | |
| Commercial assay or kit | TRIzol | Invitrogen | Cat#: 15596026 | |
| Commercial assay or kit | Direct-zol RNA MiniPrep Kits | Zymo Research | Cat#: R2051 | |
| Commercial assay or kit | RevertAid First Strand cDNA synthesis Kit | Thermo Fisher Scientific | Cat#: K1622 | |
| Commercial assay or kit | SYBR Green I Master | Roche | Cat#: 04707516001 | |
| Software, algorithm | Skewer | *Jiang et al., 2014* | | |
| Software, algorithm | bowtie | *Langmead et al., 2009* | | |
| Software, algorithm | DESeq | *Love et al., 2014* | | |
| Software, algorithm | Xtail | *Xiao et al., 2016* | | |
| Software, algorithm | DAVID | *Huang et al., 2007* | | |
| Software, algorithm | Pathway Tools | *Karp et al., 2021* | | |
| Software, algorithm | ImageQuant | GE Healthcare | | |
| Software, algorithm | Kaleidagraph | Synergy software | | |
| Software, algorithm | ImageJ | NIH | | |
| Other | MOPS EZ Rich Defined | Teknova | Cat#: M2105 | Growth medium |

## Construction of *trmD-deg* and *trmD-cont* strains

*E. coli trmD-deg* strain was constructed as described (*Carr et al., 2012*). The degron tag was amplified from the template DNA (provided by Dr Sean Moore), which encodes a FLAG tag, a His$_6$ tag, and the degron sequence YALAA followed by the promoter and the coding sequence of the Tet resistance gene. This region was amplified with primers homologous to the 3'-end and flanking sequence of the chromosomal *trmD* locus. The PCR product was purified and electroporated into competent cells of the recombinogenic *E. coli* strain SM1405 expressing λRed recombinase (*Datsenko and Wanner, 2000*). Recombination was confirmed by PCR analysis of colonies on plates containing Tet using primers homologous to the 3'-end of the chromosomal *trmD* locus. The P1 lysate of the confirmed *trmD-deg* strain was used for transduction into the recipient *E. coli* G78 strain whose chromosomal *clpX* was deleted (*Carr et al., 2012*). A resultant P1 transductant was selected on plates with Tet, confirmed by PCR, and transformed with a library of the plasmid p*clpPX* harboring a cassette encoding *clpP* and *clpX* and random mutations at the promoter and SD region that control expression of the cassette (*Carr et al., 2012*). Transformants were screened on LB + Ara (0.2%) plates to turn on expression of the cassette and to identify the clone with the highest efficiency of the degron activity as indicated by Western blots (*Figure 1C*). *E. coli trmD-cont* strain was created similarly, except that the degron tag coding for the YALAA sequence was placed after the stop codon of *trmD*. The P1 transductant harboring *trmD-cont* in the G78 strain was transformed with the p*clpPX* plasmid that showed the highest degron activity, generating the *trmD-cont* control strain. The primer sequences are shown below (5' to 3'):

> Forward recombination:
> CGGAACACGCACAACAGCAACATAAACATGATGGGATGGCGGGTGGCTCCGAC
> TACAAGG
> Reverse recombination:
> ATAATTTAATCTCTTATCCTGGGTAAACTGATATCTCGGGGGCTTAGGTCGAGGTGGCCC
> Forward confirmation at the *trmD* locus:
> ATGTGGATTGGCATAATTAGCCTGTTTCC
> Reverse confirmation at the *trmD* locus:
> GAATTCCGGTTACGAATAGCGATAACCACGCC

## Growth conditions

*E. coli* G78 strains of *trmD-deg* and *trmD-cont* were grown in LB + Glc at 37°C overnight, inoculated into 20 mL of fresh LB + Ara at 1:100, and grown for 2 hr at 37°C into the start of the log phase. This growth cycle was repeated three times. In the first cycle, cells were inoculated into 20 mL of LB + Ara (0.2%) and 10 mM Ser at OD$_{600}$ of 0.1 and grown for 1 hr at 37°C to OD$_{600}$ of ~0.4, at which point TrmD protein was no longer detected by Western blots. In the second cycle, cells were inoculated into 100 mL of LB + Ara (0.2%) and 10 mM Ser at OD$_{600}$ of 0.1 and grown for 1 hr at 37°C to OD$_{600}$ of ~0.4. In the third cycle, cells were inoculated into 500 mL at OD$_{600}$ of 0.1 and grown for 2–3 hr at 37°C to OD$_{600}$ of ~0.3. Cells of 200–300 mL were harvested by rapid filtration (see below). Strains of *trmD-KO* (*trm5–*) and *trmD-WT* (*trm5–*) were grown in LB + Ara (0.2%) at 37°C overnight and were then taken through three cycles of growth in MOPS Medium (EZ Rich Defined, Teknova) containing Glc as the only carbon source. In the first cycle, cells in the overnight culture were inoculated into 10 mL of MOPS at OD$_{600}$ of 0.1 and grown for 4 hr at 37°C to turn off the Ara-dependent expression of *trm5* and to deplete m$^1$G37-tRNAs. In the second cycle, cells were inoculated into 25 mL of MOPS + Glc and grown for 3 hr, and in the third cycle, cells were inoculated into 300 mL of MOPS + Glc and grown 2–4 hr at 37°C until OD$_{600}$ of ~0.3. Cells of 200–300 mL were harvested by rapid filtration (see below).

## Western blots

*E. coli* G78 strains of *trmD-deg* and *trmD-cont* were grown overnight in LB and then diluted 1:100 into fresh LB + Ara (0.2%). Cells were grown at 37°C and were lysed 0, 15, 30, 45, 60, and 90 min after inoculation. Cell lysate of 15 µg of protein was separated on 12% SDS-PAGE and levels of TrmD were probed by an anti-TrmD primary antibody (provided by Dr Glenn Bjork) at a dilution of 1:10,000 and a secondary anti-rabbit IgG antibody (Sigma-Aldrich) at a dilution of 1:160,000. Signals

were detected by the SuperSignal West Pico Chemiluminescent Substrate (Thermo Fisher Scientific in the Chemi-Doc XRS+ System [Bio-Rad]).

## Primer-extension assays

Primer-extension analyses of *E. coli* tRNA$^{Leu}$(CAG) were performed as described (*Masuda et al., 2019*) on *trmD-deg* and *trmD-cont* lysates generated from the same volume of cells collected at the indicated time points after switching to LB + Ara. A DNA primer targeting nucleotides 40–54 of the tRNA was chemically synthesized, [$^{32}$P]-labeled at the 5'-end by T4 polynucleotide kinase, annealed to the tRNA, and extended by Superscript III reverse transcriptase (Invitrogen) at 200 U/μL with 6 μM each dNTP in 50 mM Tris-HCl, pH 8.3, 3 mM MgCl$_2$, 75 mM KCl, and 1 mM dithiothreitol (DTT) at 55°C for 30 min. The reaction was quenched with 10 mM EDTA at 70°C for 15 min. cDNA products were separated by 12% PAGE/7 M urea and visualized by phosphorimaging. The extension product of the read-through cDNA is 53 nt in length, whereas the extension inhibition product is 15 nt in length.

## Cell harvesting and lysis

We used two different cell harvesting strategies to arrest ribosomes and block translation after cell lysis. The strategy of cell harvesting can influence the quality of ribosome profiling data (*Mohammad et al., 2019*). Cultures of *trmD-deg* and *trmD-cont* were harvested by rapid filtration on a Kontes 99 mm filtration apparatus and 0.45 μm nitrocellulose filter (Whatman) and the cells were flash-frozen in liquid nitrogen. Cells were lysed in lysis buffer (20 mM Tris-HCl, pH 8.0, 10 mM MgCl$_2$, 100 mM NH$_4$Cl, 5 mM CaCl$_2$, 100 U/mL RNase-free DNase I, and 1 mM chloramphenicol). Due to sequence-specific inhibition by chloramphenicol (*Marks et al., 2016*; *Mohammad et al., 2016*; *Nakahigashi et al., 2014*; *Orelle et al., 2013*), this cell harvesting strategy was more prone to artifacts at the codon level, creating apparent pauses when the smaller amino acids Ala, Gly, and Ser were at the penultimate position in nascent polypeptide chains. Furthermore, chloramphenicol does not fully arrest translation in cell lysates, blurring the signal (*Mohammad et al., 2019*). In contrast, cultures of *trmD-KO* (*trm5–*) and *trmD-WT* (*trm5–*), while also harvested by rapid filtration and flash-freezing, were lysed in a buffer lacking chloramphenicol but containing high concentrations of MgCl$_2$ (20 mM Tris-HCl, pH 8.0, 150 mM MgCl$_2$, 100 mM NH$_4$Cl, 5 mM CaCl$_2$, 0.4% Triton X-100, 0.1% NP-40, and 100 U/mL RNase-free DNase I). In this lysis buffer, the high MgCl$_2$ inhibits translation by preventing ribosomal conformational changes required for elongation (*Mohammad et al., 2019*). After lysis, the samples in the high MgCl$_2$ buffer were centrifuged over a sucrose cushion to collect polysomes, which were resuspended in the standard lysis buffer without the high salt to allow micrococcal nuclease (MNase) digestion and isolation of ribosome-protected mRNA fragments for deep sequencing analysis.

## Library preparation

A 10–54% sucrose density gradient was prepared in the Gradient Master 108 (Biocomp) in the gradient buffer (20 mM Tris-HCl, pH 8.0, 10 mM MgCl$_2$, 100 mM NH$_4$Cl, and 2 mM DTT). A cell lysate of 5–20 AU was loaded onto the sucrose density gradient and centrifuged in an SW41 rotor at 35,000 rpm for 2.5 hr at 4°C. Fractionation was performed on a Piston Gradient Fractionator (Biocomp). Libraries for ribosome profiling and RNA-seq were prepared as described (*Mohammad et al., 2016*; *Woolstenhulme et al., 2015*) on RNA fragments 15–45 nt in length, with the exception of RNA-seq libraries for the *trmD-KO* (*trm5–*) and *trmD-WT* (*trm5–*) samples, which were prepared using the Tru-Seq Stranded Total RNA kit (Illumina). The libraries were analyzed on a high-sensitivity BioAnalyzer (Agilent) and sequenced on the HiSeq2500 Illumina instrument.

## Analysis of sequencing data

The adaptor sequence CTGTAGGCACCATCAATAGATCGGAAGAGCACACGTCTGAA-CTCCAG TCA was removed by Skewer version 0.2.2 (*Jiang et al., 2014*). After reads mapping onto tRNA and rRNA were removed, the remaining reads were aligned to *E. coli* MG1655 genome NC_000913.2 using bowtie version 1.1.2, requiring unique mapping sites and allowing two mismatches (*Langmead et al., 2009*). The position of the ribosome was assigned using the 3'-end of reads. Reads 10–40 nt in length were included in analyses unless otherwise specified for all samples except

for RNA-seq samples from *trmD-KO* (*trm5–*) and *trmD-WT* (*trm5–*) prepared using the TruSeq kit, which were 50 nt in length.

For calculating pause scores, only reads 24–40 nt in length were used in *trmD-KO* (*trm5–*) and *trmD-WT* (*trm5–*) ribosome profiling samples because there was uncertainty about the position of the ribosome on reads shorter than 24 nt due to the different lysis buffer. In calculating pause scores, we only included genes with more than 0.1 reads per codon on average. The first seven and last seven codons in each ORF were ignored. For each instance of a codon of interest, we calculated a pause score by taking the density at the first nt only in the A site codon and dividing it by the mean density on the ORF. The average pause scores reported (*Figure 2*) represent the mean of the scores for all instances (typically thousands) of the codon of interest.

For analyses of changes in gene expression, we calculated for each gene the density of the RNA-seq or Ribo-seq in units of RPKM and then used the DESeq and Xtail packages in R with two replicates of the *trmD-deg* and *trmD-cont* samples to compute the $\log_2$ fold change in expression and the $-\log(p_{adj})$ value shown in *Figure 5*. Enrichment for GO terms was determined using the functional annotation tools in DAVID at david.ncifcrf.gov (*Huang et al., 2007*). RNA-seq data were mapped onto metabolic pathways using the Pathway Collage tool at ecocyc.org (*Karp et al., 2021*; *Keseler et al., 2017*).

## Aminoacylation of tRNA *in vitro*

Each tRNA was aminoacylated with the cognate amino acid by the cognate aaRS enzyme that had been over-expressed in BL21 (DE3) and purified via binding to a Ni-NTA resin and elution with imidazole (*Zhang et al., 2006*). Each tRNA was heat-denatured at 85°C for 3 min and re-annealed at 37°C for 15 min. Aminoacylation in steady-state conditions was performed at 37°C in a 30 μL reaction of 0.25–20 μM tRNA, 5 nM aaRS, and 20 μM [³H]-amino acid (Perkin Elmer, 7.5 Ci/mmol) in the aminoacylation buffer of 20 mM KCl, 10 mM MgCl₂, 4 mM DTT, 0.2 mg/mL bovine serum albumin, 2 mM ATP (pH 8.0), and 50 mM Tris-HCl, pH 7.5 (*Liu et al., 2011*). Reaction aliquots of 5 μL were removed at different time intervals and precipitated with 5% (w/v) trichloroacetic acid (TCA) on filter pads for 10 min twice. Filter pads were washed with 95% ethanol twice, with ether once, air-dried, and measured for radioactivity in Tri-Carb 4910 TR scintillation counter (Beckman). Counts were converted to pmoles using the specific activity of the [³H]-amino acid after correcting for signal quenching by filter pads. Data corresponding to the initial rate of aminoacylation as a function of the tRNA substrate concentration were fit to the Michaelis-Menten equation to derive the $K_m$ (tRNA), $k_{cat}$ (catalytic turnover of the enzyme), and $k_{cat}/K_m$ (tRNA) (the catalytic efficiency of aminoacylation).

## Peptide-bond formation assays *in vitro*

Peptide-bond formation assays were performed on mRNAs that varied in the second codon position (shown as XXX below) but maintained all other nucleotides, including the SD sequence (underlined) and the AUG start codon (bold face).

5'-GGGA<u>AGGAGG</u>UAAAA**AUG**-XXX-CGU-UCU-AAG-(CAC)₇-3'

Each reaction was carried out in 50 mM Tris-HCl, pH 7.5, 70 mM NH₄Cl, 30 mM KCl, 3.5 mM MgCl₂, 1 mM DTT, and 0.5 mM spermidine at 20°C as described (*Gamper et al., 2015a*; *Liu et al., 2011*). *E. coli* 70S ICs were formed by incubating 70S ribosomes, mRNA, [³⁵S]-fMet-tRNA^fMet, and initiation factors (IFs) 1, 2, and 3 with GTP for 25 min at 37°C in the reaction buffer. Each *E. coli* tRNA (4 μM) was aminoacylated with the cognate amino acid by the cognate aaRS (1 μM) to rapidly achieve plateau level of aminoacylation as confirmed by enzymatic assays. To form a TC, the aminoacylation reaction was incubated with activated EF-Tu-GTP, which was formed separately for 15 min at 37°C, and incubated in an ice bath for 15 min. To monitor peptide-bond formation, 70S ICs templated with an mRNA were mixed with a TC in an RQF-3 Kintek chemical quench apparatus. Final concentrations in each reaction (30 μL) were 0.1 μM for the 70S IC; 1.0 μM for mRNA; 0.5 μM each for IFs 1, 2, and 3; 0.25 μM for [³⁵S]-fMet-tRNA^fMet; 1.0 μM for EF-Tu for each aa-tRNA; 0.2 μM each for the aa-tRNAs; and 1 mM for GTP. Reactions were conducted at 20°C and were quenched by adding 60 μL of KOH to 0.5 M. After a brief incubation at 37°C in the quench (90 μL), aliquots of 1.8 μL were spotted onto a cellulose-backed plastic thin layer chromatography sheet and electrophoresed at 1000 V in PYRAC buffer (62 mM pyridine, 3.48 M acetic acid, pH 2.7) until the marker dye bromophenol blue (BPB) reached the water-oil interface at the anode. The position of the origin was

adjusted to maximize separation of the expected oligopeptide products. The separation of unreacted [$^{35}$S]-fMet from each of the [$^{35}$S]-fMet-peptide products was visualized by phosphor-imaging and quantified by ImageQuant (GE Healthcare) and kinetic plots were fitted using Kaleidagraph (Synergy software).

## Acid-urea gel and Northern analysis

Total RNA was extracted from cells in an acidic condition (pH 4.5) to maintain aa-tRNA levels (*Parker et al., 2020*). Cell cultures at the appropriate OD were mixed at a 1:1 vol ratio with 10% TCA and incubated on ice for 10 min. Cells were spun down at 8000 rpm (=7600× *g*) for 5 min at 4°C, resuspended in the extraction buffer (0.3 M NaOAc, pH 4.5, and 10 mM EDTA), and mixed with an equal volume of ice-cold phenol-chloroform-isoamyl alcohol (83:17:3.4), pH 5.0. The mixture was vortexed for five cycles of 1 min vortex and 1 min rest on ice, and spun at 12,000 rpm (=13,500× *g*) for 10 min at 4°C. Samples with *proS* over-expression were extracted one more time with phenol-chloroform-isoamyl alcohol (83:17:3.4) pH 5.0 for five cycles of 1 min vortex and 1 min rest on ice. The aqueous phase was separated by spinning at 12,000 rpm (=13,500× *g*) for 10 min at 4°C, and was mixed with an equal volume of cold isopropanol and incubated at −20°C for 30 min for precipitation. The RNA-containing pellet was collected after centrifugation at 14,000 rpm for 20 min at 4°C, washed with 70% ethanol in 30% 10 mM NaOAc, pH 4.5, dried, and dissolved in the gel-loading buffer (10 mM NaOAc, pH 4.5, and 1 mM EDTA). The RNA of *proS* over-expression samples dissolved in the gel-loading buffer was incubated with Proteinase K (V302B, Promega) at the final protease concentration of 0.1 mg/mL on ice for 5 min. Due to the high lability of the prolyl linkage in Pro-tRNA$^{Pro}$ to hydrolysis (*Peacock et al., 2014*), total RNA was immediately separated on an acid-urea gel. Assuming that tRNA accounts for 10% of total RNA, we made calculation to load 700 ng of total tRNA of each sample, mixed with two volumes of the acid-urea loading buffer (0.1 M NaOAc, pH 5.0, 9 M urea, 0.05% BPB, and 0.05% xylene cyanol [XC]). An intermediate size of acid-urea gel (14 × 17 cm) was used to resolve Pro-tRNA$^{Pro}$ from uncharged tRNA$^{Pro}$, whereas a Bio-Rad standard size mini gel was used to resolve Arg-tRNA$^{Arg}$ and Tyr-tRNA$^{Tyr}$ from uncharged tRNA. Each acid-urea gel was made with 6.5% polyacrylamide, 7 M urea, and 0.1 M NaOAc, pH 5.0, and run at 250 V for 3 hr 45 min at 4°C for the intermediate size gel and at 100 V for 2 hr 30 min for the mini gel. Following electrophoresis, the region between BPB and XC was excised and washed in 1× Tris-borate pH 8.0 and EDTA (TBE) buffer. Gels were transferred to a wetted nitrocellulose membrane in 1× TBE using Trans-Blot Turbo Transfer System (1704150, Bio-Rad) at constant 25 V for 20 min. The electroblotted membranes were briefly air-dried before crosslinking the RNA on the membrane using an 'optimal crosslink' in a UV crosliker (FB-UVXL-1000, Thermo Fisher Scientific). The membranes were pre-incubated in a hybridization buffer (0.9 M NaCl, 90 mM Tris-HCl, pH 7.5, 6 mM EDTA, 0.3% SDS, and 1% dry milk) at 37°C for 1 hr. Three DNA oligonucleotide probes, one targeting positions 18–36 of *E. coli* tRNA$^{Pro}$(UGG) (5'-CCAAACCAGTTGCGCTACCA-3'), one targeting positions 19–36 of *E. coli* tRNA$^{Arg}$(CCG) (5'-CGGAGGGCAGCGCTCTAT-3'), and the third targeting positions 18–36 of *E. coli* tRNA$^{Tyr}$(QUA) (5'-TACAGTCTGCTCCCTTTGGC-3') were each [$^{32}$P]-labeled at the 5'-end. These probes (each at 10$^6$ cpm) were incubated with the pre-washed membrane in the hybridization buffer for 12 hr while shaking. The probes were washed off by a 2× SSC buffer (0.3 M NaCl, 30 mM Na-citrate) at 37°C two times, each time with mild shaking for 10 min. The membranes were then dried and exposed to an imaging plate overnight. Imaging analysis was performed with a phosphor-imager (Typhoon IP, GE Healthcare) and the bands were quantified using ImageJ software (NIH). The charged fraction was calculated as the area encompassing the charged band divided by the sum of the charged and uncharged bands.

## Analysis of aaRS activity in cell lysates

Harvested cells were resuspended in a lysis buffer (50 mM Tris-HCl pH 8.0 and 150 mM NaCl) and were disrupted by a Bioruptor Pico device (Diagenode) through 12 cycles of sonication and intermittent rest at 10 and 45 s, respectively. The generated cell lysate was cleared of debris by ultracentrifugation through MTX-150 with the S100-AT4 rotor (Thermo Fisher Scientific) at 500,000× *g* for 2 hr at 4°C, and was concentrated through Amicon Ultra-4 Centrifugal Filter Units (Millipore) to ~0.5 mL. The concentrated lysate was washed with 4 mL of the binding buffer (10 mM Tris-HCl pH 7.5, 30 mM KCl, 10 mM MgCl$_2$, and 1 mM DTT) and concentrated again through the same Amicon unit to

1.6 mL. To prepare for post-DEAE fraction of the lysate, two aliquots of 300 µL DEAE sepharose were washed with 1 mL water by spinning at 3000× *g* for 2 min, and two washes with 1 mL of the binding buffer each. The 1.6 mL concentrated cell lysate was divided into two, and each was mixed with the washed DEAE sepharose and incubated at 4°C with rotation for 1 hr. The slurry was transferred to a gravity column and washed with 10 volumes of 1.2 mL of the binding buffer. The aaRS-containing fraction was eluted by 4 mL of the elution buffer (10 mM Tris-HCl pH 7.5, 400 mM KCl, 10 mM MgCl$_2$, and 1 mM DTT), concentrated by two steps of Amicon Ultra-0.5 mL centrifugal filtering down to 100 µL, exchanged with the dialysis buffer (20 mM Tris-HCl pH 7.5, 130 mM KCl, 10 mM MgCl$_2$, and 1 mM DTT), and followed by dialysis in a Slide-A-Lyzer MINI Dialysis Device, 10 k MWCO (Thermo Fisher Scientific) to remove small molecules. This post-DEAE fraction was determined for protein concentration by the Bradford assay. Molar concentration of proteins in the post-DEAE cell lysate was estimated based on an average molecular weight of *E. coli* proteins at 40 kDa.

Transcripts of *E. coli* tRNA$^{Pro}$(UGG), tRNA$^{Arg}$(CCG), and tRNA$^{Tyr}$(QUA) were made by *in vitro* transcription with T7 RNAP to generate the G37-state of each. Transcripts of *E. coli* tRNA$^{Pro}$(UGG) and tRNA$^{Arg}$(CCG) were subsequently methylated with m$^1$G37 by purified recombinant *E. coli* TrmD (*Christian et al., 2004*) to generate the m$^1$G37-state of each. Intracellular concentration of ProRS, ArgRS, and TyrRS in the post-DEAE lysate was estimated assuming each representing 1% of total proteins to allow calculation of the molar concentration of each enzyme. The final concentration of each enzyme was used at 20 nM ProRS, 20 nM ArgRS, and 7 nM TyrRS in the post-DEAE fraction for aminoacylation of 200 nM tRNA in order to produce linear synthesis of aa-tRNA over time *Figure 4C*. Aminoacylation was performed with 20 µM [$^3$H]-proline, [$^3$H]-arginine, or [$^3$H]-tyrosine and quantified as above (see section 'Aminoacylation of tRNA *in vitro*').

To investigate the effect of over-expression of aaRSs, *E. coli* *trmD*-KO strain in BL21(DE3) maintained by human *trm5* (*Gamper et al., 2015a*; *Masuda et al., 2019*) was transformed with plasmid pET22b expressing *E. coli* *proS*, *argS*, or an empty vector. Overnight cultures in MOPS medium with 0.15% Ara + 0.05% Glc were inoculated 1:100 to fresh MOPS medium with 0.15% Ara + 0.05% Glc to induce the m$^1$G37+ condition, or to MOPS with 0.2% Glc to induce the m$^1$G37– condition, each harboring 0.025 mM isopropyl-β-D-thiogalactoside (IPTG) to induce expression of the plasmid-borne gene. After two cycles of growth and dilution at 37°C, cells were harvested at OD$_{600}$ = 0.3–0.5 for isolation of total RNA and for preparation of cell lysate. To assay viability on a plate, overnight cultures in LB medium supplemented with 0.2% Ara were inoculated to fresh LB without Ara and grown for 3 hr to deplete pre-existing Trm5 protein and m$^1$G37-tRNAs. After depletion, cells were made by a 10-fold serial dilution, spotted on M9 plates containing 0.025 mM IPTG, and 0.2% Ara or 0.2% Glc as the only carbon source, and grown at 37°C for overnight.

## Reporter assay

The *leuL-leuA* genes from the *leu* operon was cloned from the *E. coli* MG1655 genomic DNA into pKK223-3 by NEBuilder HiFi DNA Assembly Cloning Kit (NEB) and the first 300 bp portion of *leuA* was translationally fused with the nLuc reporter gene. The four consecutive Leu codons CUA in *leuL* were mutated into the synonymous UUA codons by inverse PCR and self-ligation. The WT (CUA) and mutant (UUA) plasmids were transformed into *E. coli* JM109 *trmD*-KO cells maintained by human *trm5* (*Demo et al., 2021*) together with pZS2R-mCherry plasmid for constitutive expression of mCherry for normalization. Cells were grown overnight in MOPS medium in the m$^1$G37+ condition and inoculated 1:100 to fresh MOPS medium containing 0.1 mM IPTG and grown in m$^1$G37+ and m$^1$G37– conditions for 6 hr at 37°C. The nLuc readout was measured using Nano-Glo Luciferase Assay System (Promega), followed by normalization by the mCherry fluorescence level.

## qPCR analysis

To test the effect of the Pro-Pro CCA-CCG codon motif on gene expression of *ilvL*, mutations were introduced to change the motif to Ala-Ala GCA-GCG on the *E. coli* chromosome of *trmD*-KO/MG1655 strain, using CRISPR-Cas9 mutagenesis (*Reisch and Prather, 2017*). WT and CRISPR-edited mutant cells were grown and harvested for total RNA extraction using TRIzol (Invitrogen) and Direct-zol RNA MiniPrep Kits (Zymo Research) per manufacturer's instructions. It was processed to remove genomic DNA using RQ1 RNase-free DNase (M6101, Promega), followed by cDNA synthesis using RevertAid First Strand cDNA synthesis Kit (K1622, Thermo Fisher Scientific) per manufacturer's

instructions. qPCR was carried out using LightCycler 96 Instrument. For each targeted gene, 18 µL of qPCR master mix containing 10 µL of SYBR Green I Master (04707516001, Roche), 7.6 µL of nuclease-free PCR grade water, 0.2 pmoles of gene-specific forward primer, and 0.2 pmoles of gene-specific reverse primer was combined with 2 µL of cDNA template into each well of a 96-well plate (LightCycler 480 Multiwell Plate 96, white, 04729692001, Roche). qPCR amplification was performed using the following cycling conditions: preincubation for 10 min at 95℃, followed by 45 cycles of three steps of amplification with 10 s at 95℃, 10 s at 48℃, and 10 s at 72℃. The transcript level for each gene was determined using the standard curve generated from the control template by plotting the Cq values (y-axis) against the log initial concentration (x-axis). Results were confirmed by normalization with idnT mRNA. Primers used are shown below (5' to 3'):

Forward primer for *ilvL*: ATGACAGCCCTTCTACG
Reverse primer for *ilvL*: CTAAGCCTTTCCTCGTC
Forward primer for *ilvG*: ATGAATGGCGCACAGTGG
Reverse primer for *ilvG*: CCTGCTCATGTCGGCAT

## Data availability

Sequencing data have been deposited in raw FASTQ files at the SRA and processed WIG files at the GEO under accession code GSE165592. Custom Python scripts used to analyze the ribosome profiling and RNA-seq data is freely available at https://github.com/greenlabjhmi/2021_TrmD (copy archived at swh:1:rev:034808ee3fd651b0cf94d091b52aff7c6955ef1c) (*Buskirk, 2021*).

## Acknowledgements

The authors thank Chris Woolstenhulme and Ryuma Matsubara for preliminary experiments, Sean Moore and Ana Carr for materials to generate the *trmD-deg* strain, and Glenn Bjork for anti-TrmD antibodies. This study was funded by a JSPS fellowship to IM, NIH grants GM134931 to YMH, and GM110113 to ARB. The funders had no role in study design, data collection, and interpretation, or the decision to submit the work for publication.

## Additional information

### Funding

| Funder | Grant reference number | Author |
|---|---|---|
| National Institute of General Medical Sciences | GM134931 | Ya-Ming Hou |
| National Institute of General Medical Sciences | GM110113 | Allen R Buskirk |

The funders had no role in study design, data collection and interpretation, or the decision to submit the work for publication.

### Author contributions

Isao Masuda, Data curation, Investigation, Methodology, Writing - review and editing; Jae-Yeon Hwang, Howard Gamper, Conceptualization, Investigation, Methodology, Writing - review and editing; Thomas Christian, Sunita Maharjan, Fuad Mohammad, Investigation, Methodology, Writing - review and editing; Allen R Buskirk, Ya-Ming Hou, Conceptualization, Supervision, Funding acquisition, Investigation, Methodology, Writing - original draft, Project administration, Writing - review and editing

### Author ORCIDs

Isao Masuda (iD) https://orcid.org/0000-0001-9385-4424
Allen R Buskirk (iD) https://orcid.org/0000-0003-2720-6896
Ya-Ming Hou (iD) https://orcid.org/0000-0001-6546-2597

Decision letter and Author response
Decision letter https://doi.org/10.7554/eLife.70619.sa1
Author response https://doi.org/10.7554/eLife.70619.sa2

## Additional files

### Supplementary files
• Source data 1. Original image files and kinetic parameters.

• Transparent reporting form

### Data availability

Sequencing data have been deposited in raw FASTQ files at the SRA and processed WIG files at the GEO under accession code GSE165592. Custom Python scripts used to analyze the ribosome profiling and RNA-seq data is freely available at https://github.com/greenlabjhmi/2021_TrmD (copy archived at https://archive.softwareheritage.org/swh:1:rev:034808ee3fd651b0cf94d091b52aff7c6955ef1c).

The following dataset was generated:

| Author(s) | Year | Dataset title | Dataset URL | Database and Identifier |
|---|---|---|---|---|
| Masuda I | 2021 | Loss of N1-methylation of G37 in tRNA induces ribosome stalling and reprograms gene expression | http://www.ncbi.nlm.nih.gov/geo/query/acc.cgi?acc=GSE165592 | NCBI Gene Expression Omnibus, GSE165592 |

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
