## [Decision Letter]

**Acceptance summary:**

This study presents a combination of biochemical, genetic and genomics experiments to show that the lack of TrmD-catalyzed modification in the anticodons of several bacterial tRNAs leads to less efficient aminoacylation of the tRNAs by the corresponding aminoacyl-tRNA synthetases. This leads to ribosome pausing when the respective codons need to be decoded, which evokes transcriptional derepression of the Leu and Ilv operons and produces other changes in the transcriptome that strongly resemble the stringent response.

**Decision letter after peer review:**

Thank you for submitting your article "Loss of N 1-Methylation of G37 in tRNA Induces Ribosome Stalling and Reprograms Gene Expression" for consideration by *eLife*. Your article has been reviewed by 3 peer reviewers, one of whom is a member of our Board of Reviewing Editors, and the evaluation has been overseen by James Manley as the Senior Editor. The reviewers have opted to remain anonymous.

Essential revisions:

1) Document the level of m1G37 modification by TrmD of the T7 tRNA transcripts employed in biochemical experiments.

2) Properly characterize the kinetic data as measuring the rates of dipeptide formation, encompassing all of its partial reactions, rather than rates of peptidyl transfer per se.

3) Provide more convincing results and additional controls for experiments in Figures 3C and E.

4) Modify the claim that this is the first report showing that a tRNA modification affects its aminoacylation.

5) In the absence of analyzing a relA mutant, try to provide direct evidence for activation of the stringent response by measuring ppGpp levels in the trmD mutant.

6) Consider revising/shortening the discussion of changes in metabolism beyond the stringent response for which there is little supportive evidence or mechanistic understanding. For example, the uncertain connection between m1G37 methylation and glycolysis and the glyoxylate cycle detracts from the main message of the study.

*Reviewer #2 (Recommendations for the authors):*

Data presentation:

1) Figure 1D, could the authors explain why trmD levels drop abruptly?

2) Figure 1E, I was curious as to why the primer-extension efficiency appears to increase as a function of cell-harvest time. I am assuming similar amounts of tRNA template was added to these reactions.

3) Figure 3C, the data on the effect of proS overexpression is not convincing. The signal is smeary and too weak to be quantified accurately. Also, proper controls to show that other tRNAs, which are not substrates for trmD are not affected by its depletion, are missing.

4) Similarly, aminoacylation reactions with non-trmD substrates are needed for figure 3E to show that equivalent amounts of lysates were added to the different reactions.

*Reviewer #3 (Recommendations for the authors):*

1) ll. 294-317. The authors rightfully mention that their 'peptidyl transfer efficiency' assay measures the overall rate of all the events at the A-site, form initial tRNA binding, to GTP hydrolysis, to accommodation and, finally, to peptide bond formation per se. Therefore, they have to be more accurate in presenting their conclusions. Thus, the statement that "loss of m1G37 only affected peptidyl transfer for one isoacceptor at one codon" or that the loss of methylation of tRNA(Arg) "reduced… peptide bond formation" seem to be misleading making reader believe that there is a direct effect of the tRNA anticodon loop methylation on the rate of the chemical reaction of peptide bond formation.

2) ll.249-261. Was the extent of in vitro methylation by TrmD of tRNA T7 transcripts experimentally assessed? If not, what is the evidence that those transcripts were sufficiently methylated?

3) ll. 481-525. I found the discussion of changes in metabolism (in comparison with those observed at stringent response) to be too excessive and less interesting than the rest of the work. There is a lot of handwaving here and the conclusions do not really contribute to the main story but rather dilute it. I recommend to either completely eliminate this section or to at least dramatically reduce its length – down to a few sentences and the key conclusions.

---

## [Author Response]

Essential revisions:1) Document the level of m1G37 modification by TrmD of the T7 tRNA transcripts employed in biochemical experiments.

We have documented the level of m^1^G37 methylation by TrmD of the T7 tRNA transcripts used in biochemical experiments. We showed that the methylation was near 100% for the transcript of *E. coli* tRNA^Pro^(UGG) and near 70% for the transcript of *E. coli* tRNA^Arg^(CCG) (Figure 3—figure supplement 1A). These results indicate that the kinetic parameters derived from these methylated tRNAs are meaningful.

2) Properly characterize the kinetic data as measuring the rates of dipeptide formation, encompassing all of its partial reactions, rather than rates of peptidyl transfer per se.

We agree that our measurements of dipeptide formation encompass all of its partial reaction and that the rate constant *k*_obs_ represents a composite term, not just peptidyl transfer per se. We have provided more details to our measurements of *k*_obs_, which is the slope of the linear synthesis of product vs. time under a substrate-limiting condition (pp13-14). Raw data for the linear dipeptide formation over time are shown for *E. coli* tRNA^Pro^(UGG) against the CCA codon (Figure 3—figure supplement 1B, 1C) and for *E. coli* tRNA^Arg^(CCG) against the CGG codon (Figure 3—figure supplement 1D, 1E).

3) Provide more convincing results and additional controls for experiments in Figures 3C and E.

We thank the reviewers for this request. We have re-organized our data on aminoacylation and moved the previous Figures 3C and 3E to the revised Figures 4A and 4C, respectively.

In the revised Figure 4A, we provided new results that clearly resolve the charged Pro-tRNA^Pro^ from the uncharged tRNA^Pro^ by acid-urea gels (Figure 4A, left panel). We achieved this clear resolution by an extra phenol/chloroform extraction of total RNA from cells, followed by incubation of the RNA sample with proteinase K at 0.1 mg/mL for 5 min on ice to remove residual proteins. The RNA was then run on an acid-urea gel and probed for *E. coli* tRNA^Pro^(UGG). The results showed that the level of aminoacylation is decreased from 84% in the m^1^G37-abundant condition to 68% in the m^1^G37-deficient condition, but that it is restored to 82% in the m^1^G37-deficient condition upon over-expression of *proS*. These results support the notion that m^1^G37 deficiency reduces aminoacylation of tRNA^Pro^(UGG) in vivo, and that the reduction can be overcome by over-expression of *proS*,

In the revised Figure 4A, we also included the acid-urea gel analysis of *E. coli* tRNA^Tyr^(QUA, Q = queuosine) as an m^1^G37-independent control, which contains ms^2^i^6^A37 (2-methyl-thio-*N*^6^-isopentenyl adenosine) and is not methylated by TrmD. The results showed that the level of aminoacylation of this tRNA is virtually identical between the m^1^G37-abundant condition (78%) and the m^1^G37-deficient condition (81%) (Figure 4A, right panel), supporting the notion that loss of m^1^G37 does not affect the aminoacylation of an m^1^G37-independent tRNA in cells.

4) Modify the claim that this is the first report showing that a tRNA modification affects its aminoacylation.

We agree with the reviewers and have removed the claim from the Discussion section (pp 24-25). We apologize for our previous over-statement. We realized that we had published a paper showing that loss of the s^2^ group from the cmnm^5^s^2^U34 (5-carboxy-amino-methyl-2-thio-uridine 34)-state in *E. coli* tRNA^Gln^ also led to reduced aminoacylation and reduced binding and accommodation to the A site (Rodriguez-Hernandez *et al.,* 2013). We have cited our tRNA^Gln^ paper in Discussion.

5) In the absence of analyzing a relA mutant, try to provide direct evidence for activation of the stringent response by measuring ppGpp levels in the trmD mutant.

We agree that measuring ppGpp levels in the *trmD* mutant would strengthen the conclusion that m^1^G37 deficiency activates the stringent response. However, this measurement is difficult to perform for several reasons.

1) The measurement is most commonly performed with culturing bacterial cells in the presence of 32P-H3PO4 in the growth media (100 mCi/mL) to determine the cellular synthesis of 32P-GTP, 32P-ppGpp, and 32P-(p)ppGpp by TLC. However, we do not have the radiation permit to work with 32P isotope in our culturing equipment, which is located in a common equipment corridor that is shared by all labs on our floor.

2) The high dosage of 32P throughout the experiments is dangerous and causing contamination all over the labs. Our students and staff members do not wish to work with this high dosage of 32P, which also raises a serious concern of our Radiation office in the University.

3) Although there are a few reports of an HPLC-MS (mass spectrometry) method, this method is not reliable, based on our extensive discussion with Richard Gourse (U. Wisconsin), an expert of the stringent response who uses the 32P assay in a dedicated apparatus located in his own lab. Specifically, the HPLC-MS method uses HPLC to separate all cellular nucleotides, followed by MS analysis of ppGpp. However, the published reports did not provide details on the MS analysis and requires purchasing new and expensive columns. Further, we have not found an MS facility in the US that has analyzed ppGpp successfully. The only MS facility that has done it is in U. Wisconsin, which is currently not working according to a discussion with Jun Wang (U. Wisconsin), another expert of the stringent response.

6) Consider revising/shortening the discussion of changes in metabolism beyond the stringent response for which there is little supportive evidence or mechanistic understanding. For example, the uncertain connection between m1G37 methylation and glycolysis and the glyoxylate cycle detracts from the main message of the study.

We agree with the reviewers. We have substantially shortened the discussion on metabolism, particularly on glycolysis and the glyoxylate cycle (pp21-22). We have also removed a previous figure that illustrated metabolic changes.

Reviewer #2 (Recommendations for the authors):Data presentation:1) Figure 1D, could the authors explain why trmD levels drop abruptly?

We thank the reviewer for this comment. We speculate that the rapid decrease of TrmD level upon turning on the *clpXP* machinery in the *trmD-degron* strain is because of the time requirement for expression of the degradation machinery to a threshold level, which once achieved would rapidly degrade TrmD protein. A similar pattern of rapid degradation of the target protein in a separate degron strain has been reported (Carr et al., 2012). Additionally, we speculate that the *clpXP* expression plasmid, which has the strongest expression strength as selected from a library of plasmids varying in Shine-Dalgarno sequences (Materials and methods), could also contribute to the drastic degradation of TrmD. These possibilities are discussed in Results “*E. coli* strains with conditional m^1^G37 deficiency” (p7).

2) Figure 1E, I was curious as to why the primer-extension efficiency appears to increase as a function of cell-harvest time. I am assuming similar amounts of tRNA template was added to these reactions.

The reviewer raised an important question. However, we did not load similar amounts of tRNA template to each primer-extension reaction. Instead, we used same amounts of cell culture for extraction of tRNA in different growth conditions. Due to the increased cell density from 0-3 h, an increased tRNA amount was produced, leading to an increased primer-extension stop at the 15 nt-band relative to the primer position in the *trmD-cont* cells. The primer is always added in excess and only the primer extension products are used in the quantification. This information is provided in the legend to the revised Figure 1D (p47).

3) Figure 3C, the data on the effect of proS overexpression is not convincing. The signal is smeary and too weak to be quantified accurately. Also, proper controls to show that other tRNAs, which are not substrates for trmD are not affected by its depletion, are missing.

Yes, we agree with the reviewer that the data on the effect of *proS* over-expression is not convincing in the previous submission. We have provided new data in the revised Figure 4A, showing clear separation of the charged Pro-tRNA^Pro^ from the uncharged tRNA^Pro^ and clear demonstration of the recovery of Pro-tRNA^Pro^ levels upon over-expression of *proS* (Figure 4A, left panel).

We have provided data for *E. coli* tRNA^Tyr^(QUA), an m^1^G37-independent tRNA, showing that its aminoacylation level is unchanged between the m^1^G37-abundant and m^1^G37-deficient conditions (*Figure 4A, right panel*). The new data for *E. coli* tRNA^Tyr^(QUA) are described (pp15-16).

4) Similarly, aminoacylation reactions with non-trmD substrates are needed for figure 3E to show that equivalent amounts of lysates were added to the different reactions.

We agree with the reviewer on this control. We have provided data for *E. coli* tRNA^Tyr^(QUA), showing that it is aminoacylated similarly by lysates prepared from m^1^G37-abundant and m^1^G37-deficient conditions (Figure 4C, right panel). This control reaction demonstrates that equivalent lysates were added to the different reactions.

Reviewer #3 (Recommendations for the authors):1) ll. 294-317. The authors rightfully mention that their 'peptidyl transfer efficiency' assay measures the overall rate of all the events at the A-site, form initial tRNA binding, to GTP hydrolysis, to accommodation and, finally, to peptide bond formation per se. Therefore, they have to be more accurate in presenting their conclusions. Thus, the statement that "loss of m1G37 only affected peptidyl transfer for one isoacceptor at one codon" or that the loss of methylation of tRNA(Arg) "reduced… peptide bond formation" seem to be misleading making reader believe that there is a direct effect of the tRNA anticodon loop methylation on the rate of the chemical reaction of peptide bond formation.

We thank the reviewer for this valuable comment. We have revised the text to show that our measurement evaluates the overall reaction at the A site, producing *k*_obs_ as a composite kinetic parameter that encompasses all of the reaction at the A site. In addition, we have provided details on how we obtained the *k*_obs_ value, i.e., it is the slope of the linear production of peptide-bond formation over time under conditions of limiting concentrations of the ribosome and substrate tRNAs, such that *k*_obs_ represented the catalytic efficiency of the overall reaction of peptide-bond formation. The details of our description are shown in Results “Reduced aminoacylation and A-site peptide-bond formation of m^1^G37-deficient tRNAs”.

2) ll.249-261. Was the extent of in vitro methylation by TrmD of tRNA T7 transcripts experimentally assessed? If not, what is the evidence that those transcripts were sufficiently methylated?

We agree with the reviewer and have provided data showing high levels of m^1^G37 methylation in vitro. These data show that m^1^G37 methylation of the transcript of *E. coli* tRNA^Pro^(UGG) reaches nearly 100% and that of the transcript of *E. coli* tRNA^Arg^(CCG) reaches nearly 70% (Figure 3—figure supplement 1A).

3) ll. 481-525. I found the discussion of changes in metabolism (in comparison ith those observed at stringent response) to be too excessive and less interesting than the rest of the work. There is a lot of handwaving here and the conclusions do not really contribute to the main story but rather dilute it. I recommend to either completely eliminate this section or to at least dramatically reduce its length – down to few sentences and the key conclusions.

We agree with the reviewer and have substantially shortened the Results section “Metabolic changes” and have removed a previous figure that illustrated metabolic changes.